# miniF2F-Lean Revisited: Reviewing Limitations and Charting a Path Forward

**Azim Ospanov** [*†]
aospanov9@cse.cuhk.edu.hk

**Farzan Farnia** [†]
farnia@cse.cuhk.edu.hk

**Roozbeh Yousefzadeh** [*]
roozbeh.yz@gmail.com

## Abstract

We perform a thorough analysis of the formal and informal statements in the miniF2F benchmark from the perspective of an AI system that is tasked to participate in a math Olympiad consisting of the problems in miniF2F. In such setting, the model has to read and comprehend the problems in natural language, formalize them in Lean language, then proceed with proving the problems, and it will get credit for each problem if the formal proof corresponds to the original informal statement presented to the model. Our evaluation results reveal that the best accuracy of such pipeline can be about 36% using the SoTA models in the literature, considerably lower than the individual SoTA accuracies, 97% and 69% reported in the autoformalization and theorem proving literature. Analyzing the failure modes, we trace back a considerable portion of this drop to discrepancies between the formal and informal statements for more than half of the problems in miniF2F. We proceed with correcting all the errors, discrepancies and simplifications in formal and informal statements, and present the *miniF2F-v2* with fully verified formal and informal statements and proofs. Evaluating the full theorem proving pipeline on *miniF2F-v2* leads to the best accuracy of 70%, a significant improvement from the 40% on the original miniF2F, yet indicating considerable misalignment between the autoformalization models and theorem provers. Our deep analysis suggests that a higher quality benchmark can help the community better evaluate progress in the field of formal reasoning and also better diagnose the failure and success modes of autoformalization and theorem proving models. Our dataset is available at `https://github.com/roozbeh-yz/miniF2F_v2`.

## 1 Introduction

Automated reasoning with computers has a long and rich history [1], and with the rise of AI, it has had major advancements in the past decades, notably DeepBlue [2], AlphaGo [3], etc. Shortly after the rise of Large Language Models (LLMs), [4] showed the remarkable ability of these models to learn the language of formal verification systems such as Lean [5] and automatically prove mathematical theorems in formal language, building on prior work such as [6]. This gave rise to the subfield of Automated Theorem Proving (ATP) in the machine learning literature which has seen considerable advancements in the past few years [7]. The shared progress in this field was partly made possible by the miniF2F benchmark [8] which consists of 488 theorems in formal and informal languages drawn from prestigious mathematical competitions and Olympiads.

Writing mathematical proofs in formal language makes the verification of proofs automated and reliable, however, learning and writing the language of formal verification systems is not easy for humans nor for the LLMs. For a human, it might take 10 times longer to write a proof in formal language compared to an informal one. LLMs, on the other hand, are likely to excel at learning these

---

[*]Huawei Hong Kong Research Center
[†]Department of Computer Science & Engineering, The Chinese University of Hong Kong

39th Conference on Neural Information Processing Systems (NeurIPS 2025).

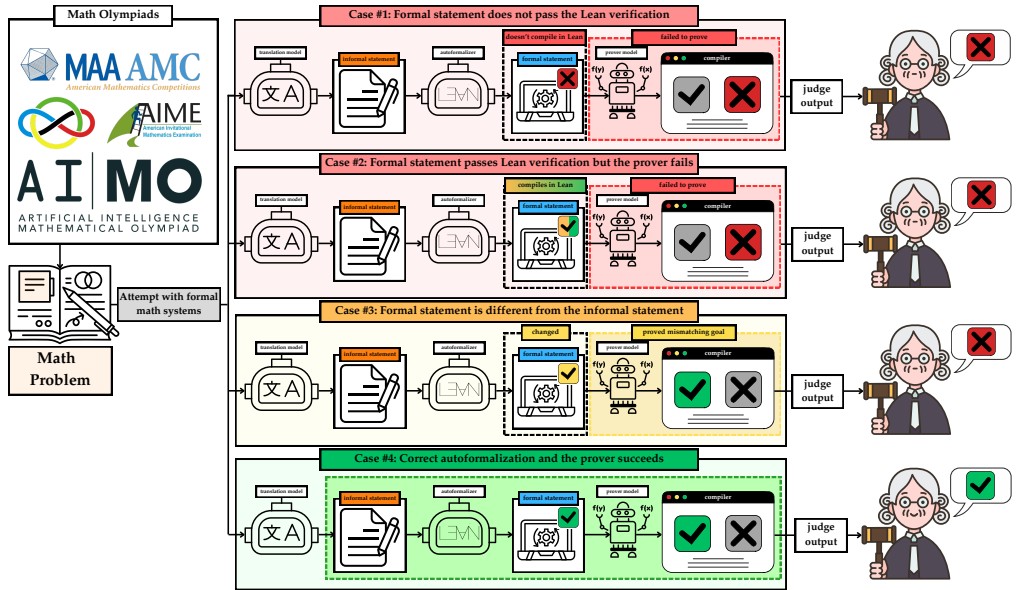

Figure 1: High-level overview of a formal math prover system operating in a Math Olympiad setting.

formal languages, and combined with other automated reasoning tools, they may help automate the process of mathematical reasoning. This has given rise to one of the main goals of the community: developing AI systems that can compete with humans in major mathematical Olympiads [9]. For example, in 2024, AlphaProof was able to reach the level of silver medal at the International Math Olympiad (IMO) [10]. Reaching that level of automated reasoning requires a model to automatically read a mathematical problem in informal language, formalize it in a formal verification language, and generate a correct formal proof that can pass the verification system.

Such automated system would fail if it changes the original problem statement and proves an altered version of the competition problem, the same way that if a human participant proves a simplified version of a problem will likely get no, or at the most, a partial credit not sufficient for a medal. Therefore, having models that can correctly translate to formal language is crucial, otherwise, there will be a need for a human in the loop to perform the formalizations. This translation, known as autoformalization in the literature, has seen major advancements on the miniF2F benchmark [11].

While the miniF2F benchmark has enabled advancements both in ATP and in autoformalization, over the years, it has also been reported to have certain limitations, such as wrong formalizations, unprovable theorems, etc. Most recently, [12] reported fixing errors in 5 theorems of miniF2F which had made them unprovable. The community on autoformalization has also reported certain inconsistencies between the formal and informal statements in miniF2F [13]. This motivates us to conduct a deep analysis of this benchmark from the holistic perspective described above. In this framework, an AI system is tasked with participating in a miniF2F competition. The system must prove all 488 theorems in the benchmark from their informal statements, with the goal of producing correct proofs for the original formulations. Figure 1 illustrates the stages at which failures can prevent an AI system from arriving at a correct solution in a Math Olympiad setting:

1. **Autoformalization failure:** The autoformalizer fails to write a formal statement that passes the formal verification compiler, e.g., Lean. In this case, the output of the autoformalizer cannot be passed to the prover, and the AI pipeline fails automatically.

2. **Translation failure:** The autoformalizer produces a formal statement that passes the compiler error-free. However, the formal statement does not match the informal one, i.e., the translation is not accurate. Such a formal statement will then be passed to the prover model. If the prover model fails to prove the problem, the pipeline automatically fails. However, if the prover model succeeds in proving such a formal statement, the judge will not give credit to the proof because it does not correspond to the original problem in the Olympiad. This is the case most prone to being neglected and inaccurately counted toward the success of AI models without a human expert examining the final proof.

3. **Prover failure:** The prover fails to prove, leading to the automatic failure of the AI pipeline. If the autoformalized statement is a correct translation of the informal one, this failure can be interpreted as a shortcoming of the prover model. However, it may be the case that the formal statement is incorrectly translated by the autoformalizer, making it unprovable or more difficult than the original problem, and such mistranslation may be the root cause of the prover's failure. In the latter case, the failure of the end-to-end pipeline may be attributed to the autoformalizer.

In this work, we build such an automated pipeline by using the SoTA models for autoformalization and for theorem proving. While the accuracy of the best autoformalization model on miniF2F is reported to be about 97% [14], and the accuracy of Kimina Prover on miniF2F is 70.8% [12], the combined accuracy of these models leads to an accuracy of 34.8% after comparing the final proofs with the original problem statements in informal language. This significant drop in accuracy comes from several sources which we will examine in detail, two of which account for most of the failures. Our extensive human evaluation of autoformalization results, across five models, reveals that their autoformalization accuracy is not as high as the ones reported in the literature, because those reported accuracies are often evaluated by LLMs and not by a human familiar with the formal language. The other reason for this major drop is that the formal statements in miniF2F, i.e., the starting point of the automated system, are often significantly simplified compared to the informal statements, and hence, when more faithful translations are given to ATPs, the theorems turn out to be more difficult, making the ATPs more likely to fail. Therefore, we observe a disconnect between the ATP literature and the autoformalization literature.

We analyze every failure mode of an end-to-end formal reasoning pipeline on the miniF2F benchmark and correct over 300 Lean 4 statements to eliminate errors and simplifications. Since our starting point is the original miniF2F, we take two steps to modify it, leading to two variations: miniF2F-v2s and miniF2F-v2c. For miniF2F-v2s, where $s$ stands for *simplified*, we only correct the mistakes in the informal statements and then modify the formal statements to exactly match the informal ones. This version is closer to miniF2F-v1, yet all theorems are correct and provable, and there are no discrepancies between the informal and formal statements. For all the problems where the formal statement contains the solution, we make sure the informal statements reflect the solutions as well.

We then take another step which leads to miniF2F-v2c where $c$ stands for *competition*. Here, we change the informal statements to reflect the exact statements in the original competitions for all the problems from IMO and AMC. If a problem has multiple choices, we keep all the choices in the informal statement and also include the choices in the formal one. Hence, the model has to first choose the correct choice and then prove it, adding an extra level of difficulty to the theorems. When original informal statement does not provide a solution and asks for the participant to first find the solution and then prove it, we also do not provide the solution in the informal and formal statements.

**Our contributions are as follows:**

- We release two corrected versions of miniF2F benchmark (both test and validation sets): simplified and competition-level. All the informal and formal statements are manually checked to exactly correspond to each other.

- We perform a thorough evaluation of autoformalization models on miniF2F-v1, v2s, and v2c, demonstrating the current shortcomings in the evaluation practices in the autoformalization literature as well as the benefits of miniF2F-v2s and v2c.

- We perform a thorough evaluation of miniF2F-v2s and v2c for the task of ATP reporting that the accuracy of SoTA models drop significantly on the problems that were excessively simplified, yet their accuracy increases on the subset of problems that were previously unprovable because of formalization errors.

- We further evaluate a complete automated pipeline of SoTA models on the task of theorem proving starting from informal statements and report their accuracy both on the original version of miniF2F and miniF2F-v2 demonstrating the better accuracy of such pipelines on miniF2F-v2.

## 2 Reviewing the miniF2F in detail

In this section, we detail various types of changes that we made in the original miniF2F benchmark for both formal and informal statements. Detailed information about the distribution of made changes can be found in Appendix H.

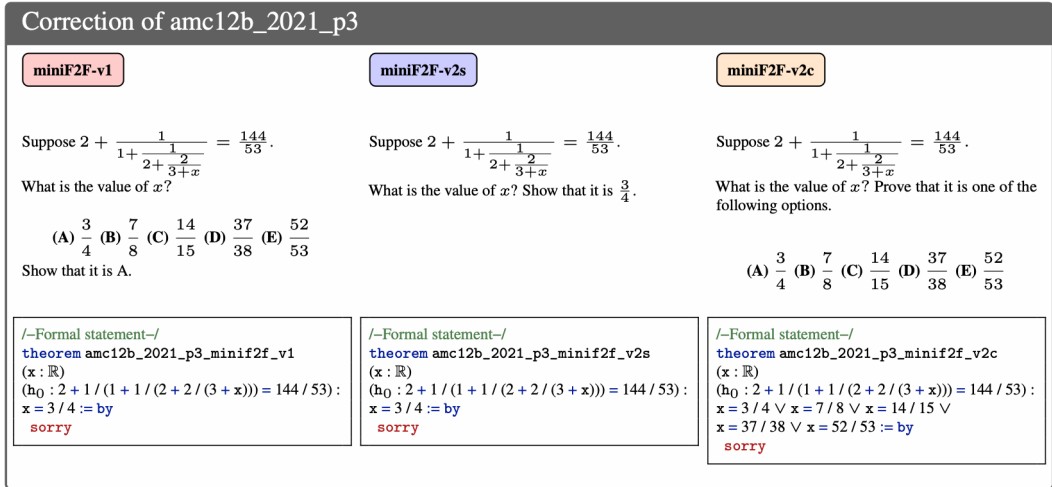

Figure 2: Correction example of amc12b_2021_p3 problem in miniF2F across versions 1, 2s and 2c.

## 2.1 Errors in informal statements

**Incomplete statements.** There are problems that do not provide enough information or do not mention the type of variables. There are also instances where the informal statement differs from the original problem statement presented in the competitions such as IMO. In all such cases, we use the original informal statements.

**Unprovable problems.** These set of problems miss critical hypotheses to prove the goal; therefore, making them unprovable.

**Multiple choices in the statement.** In miniF2F-v2s, we filtered the informal statements from multiple choice style answers to reflect only the final goal to be proved. In miniF2F-v2c, we added all the choices to the formal statements. Figure 2 illustrates an example. In v2c, we remove the correct answer from the informal statement so that it matches the original AMC statement, and introduce the multiple-choice options in the formal statement. In v2s, we omit the multiple-choice options from the informal description so that it matches the formal statement leading to a simplified problem where the solution is known. Additional examples can be found in the Appendix.

**Wrong given solution.** Another subset of incorrect informal statements provide a wrong or inconsistent solution that deviates from the original problem statement and solution.

**No given solution**. Some of the informal statements do not provide a solution after the problem description, yet the formal statements contain the solution. For miniF2F-v2s, we add the solution to the informal statements. For miniF2F-v2c, we remove the solution from the formal statement and provide a *sorry* for the ATP model to find and prove.

## 2.2 Errors in formal statements

**Excessively simplifying the problem by changing the goals or changing the conditions.** This subset of problems aim to prove simpler goals compared to the original informal description. These problems do not reflect the original difficulty of the problem; therefore, making the LLM's task easier. We segregate these problems into two categories: *simplified* and *excessively simplified*. The common culprits behind simplification are omitting part of the informal statement, simplifying the goals, adding helpful assumptions, wrong type declarations (e.g., proving a goal for non-negative real numbers instead of all real numbers, or using equivalences instead of functions). Excessive simplification cases are those where the goal is significantly altered.

**Not using the correct functions/expressions in Lean.** Another subset of formal errors are incorrect declaration of functions and expressions. In some cases, informal statements describe conditions, functions, etc, whereas the formalized version does not declare them. We fix these inconsistencies in our revised versions of miniF2F.

Table 1: Comparison of effective accuracy across different autoformalizers and theorem provers. Effective accuracy refers to a final accuracy after autoformalizer formalizes the problem statements and theorem prover attempts to prove them. Experiments are performed with @128 for translators and @32 for theorem provers. Numbers in blue are gains in accuracy as a result of higher quality benchmark. Numbers in red are drop in accuracy as a result of increased difficulty compared to the simplified problems in miniF2F-v1.

| | miniF2F | | Verification Setting | Theorem prover accuracy (%) | | | | |
| | ver. | split | excessively simplified proofs | Deepseek-Prover-V1.5-RL | Goedel-Prover-SFT | Kimina-Prover-Distill-7B | DeepSeek-Prover-V2-7B | Goedel-V2-7B |
|---|---|---|---|---|---|---|---|---|
| Herald translator | v1 | test | full score[3] | 34.4 | 37.7 | 41.0 | 50.8 | 54.1 |
| | v2s | | | 34.0 (-0.4) | 37.7 (0) | 42.2 (+1.2) | 47.5 (-3.3) | 53.3 (-0.8) |
| | v2c | | | 33.2 (-1.2) | 36.9 (-0.8) | 39.3 (-1.7) | 43.4 (-7.4) | 48.4 (-5.7) |
| | v1 | test | no score (Olympiad setting[4]) | 21.7 | 24.2 | 27.5 | 33.2 | 36.9 |
| | v2s | | | 31.1 (+9.4) | 34.8 (+10.6) | 38.9 (+11.4) | 44.7 (+11.5) | 50.0 (+13.1) |
| | v2c | | | 30.3 (+8.6) | 34.0 (+9.8) | 36.5 (+9.0) | 40.6 (+7.4) | 44.7 (+7.8) |
| | v1 | valid | full score | 43.0 | 47.1 | 50.0 | 62.7 | 64.3 |
| | v2s | | | 43.0 (0) | 46.3 (-0.8) | 50.0 (0) | 58.6 (-4.1) | 59.4 (-4.9) |
| | v2c | | | 41.4 (-1.6) | 43.4 (-3.7) | 46.3 (-3.7) | 52.9 (-9.8) | 53.3 (-11) |
| | v1 | valid | no score (Olympiad setting) | 31.6 | 34.4 | 36.9 | 44.7 | 46.7 |
| | v2s | | | 40.2 (+8.6) | 43.4 (+10) | 47.1 (+10.2) | 55.7 (+11) | 56.1 (+9.4) |
| | v2c | | | 38.5 (+6.9) | 40.6 (+6.2) | 43.4 (+6.5) | 50.0 (+5.3) | 51.2 (+4.5) |
| Kimina autoformalizer | v1 | test | full score | 42.2 | 48.0 | 60.2 | 69.7 | 77.5 |
| | v2s | | | 43.9 (+1.7) | 47.6 (-0.4) | 62.3 (+2.1) | 63.5 (-6.2) | 74.2 (-3.3) |
| | v2c | | | 40.6 (-1.6) | 46.7 (-1.3) | 56.1 (-4.1) | 55.7 (-14) | 65.6 (-11.9) |
| | v1 | test | no score (Olympiad setting) | 24.2 | 27.9 | 35.2 | 40.2 | 49.2 |
| | v2s | | | 41.8 (+17.6) | 47.5 (+19.6) | 58.6 (+23.4) | 60.2 (+20) | 69.7 (+20.5) |
| | v2c | | | 38.1 (+13.9) | 44.3 (+16.4) | 52.0 (+16.8) | 52.5 (+12.3) | 61.5 (+12.3) |
| | v1 | valid | full score | 52.0 | 55.3 | 64.8 | 75.4 | 78.7 |
| | v2s | | | 51.6 (-.4) | 52.8 (-2.5) | 65.6 (+0.8) | 72.1 (-3.3) | 74.6 (-4.1) |
| | v2c | | | 47.5 (-4.5) | 52.0 (-3.3) | 60.7 (-4.1) | 63.5 (-11.9) | 66.4 (-12.3) |
| | v1 | valid | no score (Olympiad setting) | 38.1 | 40.6 | 46.7 | 51.2 | 54.9 |
| | v2s | | | 48.0 (+9.9) | 52.0 (+11.4) | 61.1 (+14.4) | 68.4 (+17.2) | 70.1 (+15.2) |
| | v2c | | | 43.9 (+5.8) | 48.4 (+7.8) | 56.6 (+9.9) | 60.2 (+9) | 62.3 (+7.4) |

**Unprovable statements.** We identified a total of 16 problems across test and validation sets that do not have a solution in their current form, which is a critical factor when it comes to reliable evaluation of theorem provers. The errors come from improper translation of the informal statement to a formal counterpart, such as missing brackets or using a wrong function from the Mathlib library.

---

[3] In this setting, simplification is rewarded and encouraged. If a LLM, proves an excessively simplified modification of a problem, it still gets full credit. This is the setting that is perhaps inadvertently being used to evaluate the accuracy of formal reasoning models.

[4] Each Olympiad, or competition, may have different rules on giving partial scores, and these rules might even change year by year. To make our evaluation clear, we opted for the choice of giving no credit for *excessively*

Table 2: Comparison of autoformalization accuracy of Herald and Kimina translators at @128 between LLM and human evaluators. Back translation and LLM equivalence check pipeline is adopted from [14].

| Translator | Evaluator | miniF2F-v1 | | miniF2F-v2s | | miniF2F-v2c | |
|---|---|---|---|---|---|---|---|
| | | test | valid | test | valid | test | valid |
| Herald | Herald's automated judge | 97.5% | 97.1% | 96.7% | 97.5% | 95.1% | 97.1% |
| | Human | 62.7% | 69.7% | 66.0% | 68.9% | 54.1% | 60.2% |
| Kimina | Herald's automated judge | 98.4% | 98.4% | 99.6% | 98.8% | 98.4% | 98.8% |
| | Human | 90.6% | 88.1% | 91.0% | 88.1% | 75.4% | 76.6% |

# 3 Evaluation of complete formal reasoning pipelines starting from informal statements

This section presents our experiments with an end-to-end pipeline that aims to prove informal theorems with the aid of formal theorem prover systems. The setting is motivated by Math Olympiad–style problems, where the task is to produce a correct solution for a given informal statement. We selected every original problem used to construct the *miniF2F* benchmark and employed them to evaluate how well state-of-the-art autoformalizers can work with SoTA theorem provers.

For each problem we begin by feeding the informal statement to an autoformalizer; we keep the first formal output that both passes REPL verification and remains semantically faithful to the source, which is judged by human experts. We then attempt to prove the resulting goal with several theorem provers, and finally we compare the derived theorem with the original problem, recording any discrepancies. We refer to final accuracy of autoformalizer and theorem prover collaboration as "effective accuracy".

Table 1 summarizes our end-to-end results on miniF2F-v1 and miniF2F-v2. For each pair of autoformalization and theorem provers, we report two effective accuracy metrics: (i) the percentage of proofs that pass REPL verification giving credit to all proofs even for the ones that are excessively simplified compared to the original informal statements, and (ii) the percentage of proofs that both pass verification *and* align with the original problem statement, i.e., the Olympiad setting.

This table highlights the impact of having a high quality and error-free benchmark where all theorems are provable, and all the formal and informal statements match each other. For all pairs of autoformalization and theorem provers, effective accuracy is considerably higher on miniF2F-v2s and v2c compared to v1 when the pair of models are evaluated in the "Olympiad setting". This gain in accuracy is despite the fact that v2 versions of miniF2F are more difficult. In the "full score" setting, however, where the LLMs get full credit for proving excessively simplified problems, the accuracy on v2 versions of miniF2F are considerably lower. More detailed analysis of these results are explained in the following sections where we evaluate the accuracy autoformalizers and theorem provers separately on each version of miniF2F.

# 4 Evaluation of autoformalization models

This section evaluates the performance of autoformalizers on miniF2F-v1, v2s and v2c. The most notable observation of this section is that the reported accuracy of autoformalization models in the literature are largely inflated as those evaluations are typically performed by LLMs. For example, when we perform a human review of the outputs of Herald @128 that LLM has marked as 97% correct, we arrive at a much lower accuracy of 66%. And the same observations hold for Kimina autoformalizer.

We consider two specialized autoformalization systems, Herald translator [14] and Kimina autoformalizer [12], and one general-purpose model, o4-mini [15]. All experiments use a sampling budget of @1 or @128 for the dedicated autoformalizer models and @10 (with intermediate compiler

---

*simplified* proofs while giving full credit for the *simplified* proofs. In most Olympiads, an excessively simplified proof will get zero or close to zero score.

Table 3: Comparison of autoformalization accuracy of Herald translator, Kimina autoformalizer and o4-mini on the original version of miniF2F (miniF2F-v1) and miniF2F-v2. The o4-mini translation has sample budget up to @10, but the generation stops at the first correctly compiled attempt, while other models are evaluated with @1.

| miniF2F | | verified by | Model Formalization Accuracy (%) | | |
|---|---|---|---|---|---|
| ver. | split | | Herald translator | Kimina autoformalizer | o4 mini* |
| v1 | test | LLM | 49.6% | 58.6% | - |
| v1 | test | human | 55.3% | 88.1% | 46.3% |
| v2s | test | LLM | 49.2% | 54.5% | - |
| v2s | test | human | 51.6% | 79.5% | 51.6% |
| v1 | valid | LLM | 51.2% | 57.8% | - |
| v1 | valid | human | 59.4% | 82.0% | 47.1% |
| v2s | valid | LLM | 50.4% | 51.6% | - |
| v2s | valid | human | 48.0% | 78.7% | 52.5% |

feedback) for o4-mini. We note that although o4-mini has a different sample budget, we stop at the first successful compilation attempt, which puts it on par with @1 Herald translator and Kimina autoformalizer models. We exclude any autoformalization outputs that fail to pass the Lean compiler to ensure syntax correctness. Lean compiler of choice is Lean REPL [16].

To evaluate Herald translator, Gao et al. [14] used InternLM2-Math-Plus-7B [17] for back-translation and DeepSeek Chat v2.5 [18] for equivalence verification. As observed by Ye et al. [19], using an LLM as a judge reduces verification overhead but can diverge from human judgments. In our experiments, we did not change back-translation model; however we used Deepseek-V3 [20], instead of Deepseek-V2.5 for natural language validation. To present accurate results aligned with the human grasp of presented problems, we manually verified every translation and report both LLM-based and human-verified accuracies. All human verifications were conducted by Lean experts. For the @128 setting, human evaluation is performed only on the first translation that successfully passes the Lean compiler, rather than on all 128 translations.

Table 2 compares the accuracy of the LLM evaluator with human verification on both versions of miniF2F, using the Herald translator and Kimina autoformalizer with a sampling budget of @128. The LLM tends to produce false positives and therefore reports a much higher accuracy than a human evaluator. In many cases, it treats small discrepancies between statements as negligible even though they significantly affect the meaning of the statements; as shown in Section 3, these differences accumulate and reduce the practicality of full informal-to-formal pipelines. Consequently, autoformalizations should be evaluated with great care, and semantic alignment must remain strict.

To broaden our study, we conducted @1 experiments with Herald translator, Kimina autoformalizer, and o4-mini. Here, the opposite pattern appears: the LLM evaluator assigns lower accuracies than the human evaluator. We hypothesize that with a larger sampling budget the LLM has a higher chance of hallucinating and assigning incorrect labels, whereas at @1 it behaves more conservatively. Moreover, while human evaluation at @128 shows only a 10–15% improvement over @1, the LLM evaluation suggests almost double the gains.

Kimina Autoformalizer attains higher accuracy than Herald Translator under both LLM-based and human verification, with accuracies ranging from 78% to 88% across both versions of miniF2F. However, when evaluating on miniF2F-v1 against miniF2F-v2, both Herald Translator and Kimina Autoformalizer exhibit a performance decline, whereas o4-mini improves on the corrected dataset. This finding suggests that current autoformalizers may suffer from data contamination.

We also present the failure modes of each autoformalization model by topic in Appendix I. This analysis highlights the types of mathematical domains where current autoformalizers struggle the most, providing guidance for future works.

## 5   Evaluation of theorem provers on formal statements

In this section we conduct a series of experiments only with whole-proof generation LLMs starting from the formal statements in the miniF2F-v1, v2s, and v2c. This is to provide insights about the

Table 4: Comparison of whole-proof generation models' accuracy on the original version of miniF2F and miniF2F-v2.

| Dataset | Deepseek-Prover-V1.5-RL | Goedel-Prover-SFT | Kimina-Prover-Distill-7B | DeepSeek-Prover-V2-7B | Goedel-V2 |
|---------|------------------------|-------------------|--------------------------|-----------------------|-----------|
| v1-test | 50.0% | 58.2% | 65.2% | 73.4% | 82.0% |
| v2s-test | 41.0% | 48.4% | 59.0% | 68.1% | 74.2% |
| v2c-test | 38.1% | 46.3% | 57.0% | 64.4% | 72.5% |
| v1-valid | 63.9% | 68.9% | 73.0% | 79.4% | 83.6% |
| v2s-valid | 55.3% | 59.8% | 68.0% | 73.4% | 77.9% |
| v2c-valid | 52.0% | 57.8% | 67.6% | 70.5% | 73.4% |

Table 5: Comparison of whole-proof generation models' accuracy on the subset of problems in miniF2F-v2 that were previously unprovable on the original version of miniF2F.

| Unprovable subset | Deepseek-Prover-V1.5-RL | Goedel-Prover-SFT | Kimina-Prover-Distill-7B | Deepseek-Prover-V2-7B | Goedel-V2 |
|-------------------|------------------------|-------------------|--------------------------|-----------------------|-----------|
| miniF2f-v2s-test (out of 13) | 4 (30.8%) | 4 (30.8%) | 5 (38.5%) | 5 (38.5%) | 8 (61.5%) |
| miniF2f-v2s-valid (out of 3) | 1 (33.3%) | 1 (33.3%) | 1 (33.3%) | 1 (33.3%) | 1 (33.3%) |

differences in the formal statements of the two versions of miniF2F while ablating the effect of autoformalizers. Following the literature, we use a sampling budget of @32 in all runs. Deepseek-Prover-V1.5-RL [21], Goedel-Prover-SFT [22] and Deepseek-Prover-V2-7B [23] are evaluated under `Lean v4.9.0`, matching the versions used in their original papers, whereas Kimina-Prover-Distill-7B [12] is tested with the newer `Lean v4.17.0`. All experiments were performed on eight NVIDIA A5000 GPUs with 128 CPU cores.

**Performance of theorem provers on miniF2F-v2.** Table 4 reports the accuracy of selected theorem provers on miniF2F-v1, v2s, and v2c. Notably, the accuracy of every theorem prover is lower on miniF2F-v2, since many simplifications made in miniF2F were reverted back, and theorems became more challenging. The proposed dataset poses a greater challenge to state-of-the-art LLMs. We observe that increased difficulty of v2c leads to 11.2% drop in accuracy for Deepseek-Prover-V2-7B, and many failure cases come from Math Olympiad level problems that are especially hard to prove.

**Performance of theorem provers on the modified problems in miniF2F-v2.** Although the overall accuracy declines on miniF2F-v2, it is important to note that this version corrects sixteen statements that were unprovable in the original benchmark. All theorem provers can now solve a subset of these repaired problems, which raises their accuracy on this specific group of tasks. The results are reported

Table 6: Comparison of whole-proof generation models' accuracy on the subset of problems in miniF2F-v2 that were *simplified* in the original version of miniF2F.

| Simplified subset | Deepseek-Prover-V1.5-RL | Goedel-Prover-SFT | Kimina-Prover-Distill-7B | Deepseek-Prover-V2-7B | Goedel-V2 |
|-------------------|------------------------|-------------------|--------------------------|-----------------------|-----------|
| miniF2F-v1-test (out of 40) | 29 (72.5%) | 30 (75.0%) | 31 (77.5%) | 36 (90.0%) | 37 (92.5%) |
| miniF2f-v2s-test (out of 40) | 23 (57.5%) | 24 (60.0%) | 25 (62.5%) | 29 (72.5%) | 33 (82.5%) |
| miniF2F-v1-valid (out of 48) | 27 (56.2%) | 29 (60.4%) | 30 (62.5%) | 34 (70.8%) | 38 (79.2%) |
| miniF2f-v2s-valid (out of 48) | 25 (52.1%) | 25 (52.1%) | 27 (56.2%) | 31 (64.6%) | 35 (72.9%) |

Table 7: Comparison of whole-proof generation models' accuracy on the subset of problems in miniF2F-v2 that were *excessively simplified* in the original version of miniF2F.

| Excessively Simplified subset | Deepseek-Prover-V1.5-RL | Goedel-Prover-SFT | Kimina-Prover-Distill-7B | Deepseek-Prover-V2-7B | Goedel-V2 |
|---|---|---|---|---|---|
| miniF2F-v1-test (out of 45) | 21 (46.7%) | 33 (73.3%) | 33 (73.3%) | 37 (82.2%) | 42 (93.3%) |
| miniF2f-v2s-test (out of 45) | 10 (22.2%) | 13 (28.9%) | 24 (53.3%) | 27 (60.0%) | 36 (80.0%) |
| miniF2F-v1-valid (out of 36) | 26 (72.2%) | 26 (72.2%) | 28 (77.8%) | 29 (80.6%) | 31 (86.1%) |
| miniF2f-v2s-valid (out of 36) | 7 (19.4%) | 9 (25.0%) | 17 (47.2%) | 17 (47.2%) | 19 (52.8%) |

in Table 5. Since we provide the formal proofs for all problems in miniF2F-v2, one can be sure that all theorems are provable.

To further assess the impact of our revisions, we compare prover accuracy on the previously defined *simplified* and *excessively simplified* subsets. Table 6 shows that theorems in the *simplified* group pose only a modest challenge: accuracy falls for every model, yielding a 15-18 % drop on the test set, while the validation set experiences an even smaller decline. The picture changes significantly for the *excessively simplified* subset. Here, every model struggles: test-set accuracy decreases by more than 20%, and the validation set loses over 30%. Deepseek-Prover-V1.5-RL and Goedel-Prover-SFT suffer the largest losses, in some cases up to 40–50%. In contrast, Kimina-Prover-Distill-7B remains comparatively robust, proving 53.3% of the test problems and 47.2% of the validation problems, indicating strong generalization. Deepseek-Prover-V2-7B also struggles with more challenging counterparts and exhibits a 22.2% drop in test set and 33.4% drop in validation set. By making the subset of problems closer to the intended difficulty, we see that each LLM struggles at least with some of the new theorems. This indicates the importance of the renewed version of miniF2F with scaled difficulty.

## 6 Related Works

**Autonomous AI systems excelling in reasoning and scientific discovery.** With the rise of generative models, we have seen their success in scientific discoveries [24]. For example, FunSearch [25] succeeded in writing a bin-packing algorithm that is faster than any human written algorithm. These systems, usually consisting a LLM at their core, have shown remarkable ability in automated reasoning. For example, AlphaGeometry [26] was able to reach gold-medal level in solving geometry problems at IMO. AlphaProof [10], similarly reached silver-medal level in proving IMO problems in number theory and algebra. Other examples include AlphaCode [27] and AlphaEvolve [28].

Silver and Sutton [29] suggest that we will see a new generation of AI agents that will reach unprecedented abilities predominantly by learning from experience. This argument largely draws not just from recent successes of AI systems, but also looks back at the successes of systems such as AlphaGo which was able to learn the game of Go merely by playing with an automated adversary, and ultimately reaching the level of expertise to beat the human champion. When the same algorithm was transformed to play the Japanese chess, it also passed the best human performance while the model developers had no familiarity with the Japanese chess. Indeed, the strategies utilized by the model were known to fail by human champions, yet the models devised them in ways that were able to beat those same champions.

The idea here is that to make new discoveries or to arrive at models that can find better ways of playing a game or can excel at proving mathematical theorems, the models have to be given the space to explore the possibilities by themselves with no or minimal human intervention. From this perspective, a fully automated pipeline for mathematical reasoning would be preferable to a pipeline that needs a human in the loop for formalization, etc. Hence, in this work, we suggest a fully automated pipeline to evaluate AI systems for formal reasoning.

**Automated Theorem Proving.** Automatically proving mathematical theorems has a rich history including SAT and SMT solvers. In recent years, LLMs have shown a remarkable capability to generate formal proofs by themselves [30, 31, 30, 32, 33, 34, 12] or with help from other automated systems such as retrieval-based and/or search methods [35, 34, 36, 37, 38, 10, 39]. Before the release of Kimina Prover, SoTA LLMs on the task of theorem proving were only able to prove theorems with relatively short proofs in formal language [40], heavily relying on automated solvers in Lean such as nlinarith. The longest proof written by Goedel Prover on miniF2F consisted of $\sim 10$ lines. Kimina Prover, however, increased this limit to a few hundred lines using a longer context length.

**Autoformalization.** This can be viewed as a translation task [41, 42] where statements from informal language are translated to the language of a formal verification system such as Lean. Informalization is the reverse translation from formal language to an informal one which is considered an easier task. LLMs have shown a remarkable ability in translation tasks especially when large corpus of text are available in two or more languages. Similarly LLMs are good at writing code in languages such as Python and C++ [43]. We also have seen gradual improvements in the accuracy of LLMs in autoformalization [11, 34].

Herald [14], the current state-of-the-art in the literature, reports an accuracy of 97% on the miniF2F while its accuracy is measured by an LLM and not verified by a human familiar with Lean. There are generally two difficulties in the field of autoformalization. First, high quality data paired in formal and informal languages is scarce. Second, there are no automated systems that can reliably verify the correctness of a translation [44], and as we will see, LLMs may not be reliable in evaluating whether a translation is correct. Even when the ground truth is available in formal language, it may not be easy, for a human nor a LLM, to evaluate whether a freshly generated formal statement is equivalent to the ground truth. There has been considerable work in this domain trying to automate the evaluation of equivalent formal statements [45, 46].

**Benchmarks for formal reasoning.** Over the past few years, the community has introduced several formal-mathematics benchmarks of varying difficulty. ProofNet [47] focuses on undergraduate-level mathematics, while PutnamBench [48] collects problems from the William Lowell Putnam Competition (1962–2023). NuminaMath [49] and miniCTX [50] target advanced theorems drawn from Lean projects and textbooks. Recently proposed, Con-NF [44] is specifically designed for an autoformalization task. In this work, we concentrate on the miniF2F dataset [51], which consists of 488 problems sourced from textbooks and competitions such as IMO, AIME, and AMC.

**Reliability of mathematical benchmarks.** Ensuring the reliability of LLM benchmarks is a critical concern for the research community. To accurately assess model capabilities, benchmarks must be both comprehensive and error-free. Vendrow et al. [52] evaluated numerous datasets across various tasks and introduced the concept of *platinum* benchmarks, i.e. those containing minimal errors and verified by human experts. The integrity of these benchmarks is essential for measuring progress in formal reasoning. Accordingly, we dedicate our efforts to exhaustively verify the miniF2F dataset.

## 7 Limitations

This work only covers Lean language, even though miniF2F is available for other languages, too. Our corrections to the informal statements are applicable to all users of miniF2F.

## 8 Conclusion

We introduced *miniF2F-v2*, a revised version of the miniF2F benchmark. Hundreds of theorems were re-verified and changed to match the difficulty of their source problems, and the sixteen unprovable statements in the original dataset were fixed. To recreate a realistic setting of math Olympiad competitions, using the SoTA models in the literature, we built a completely automated pipeline of theorem proving starting from natural language statements. Our evaluation results show that in such setting the accuracy of current models will significantly drop on miniF2F-v1. However, when we do the same evaluation on miniF2F-v2, some of the lost accuracy is gained back because of the higher quality of the revised dataset. We further compared LLM evaluation of the outputs of autoformalization models with expert human verification and observed a substantial gap: LLMs marked many formalizations as correct even though they differed from the intended statements. By our evaluation, the accuracy of SoTA autoformalization model on miniF2F-v1 is 66%, not the reported 97%. We hope that *miniF2F-v2* will serve as a clearer and more demanding benchmark and will guide future progress in both autoformalization and formal theorem proving.

## Acknowledgments

The work of Farzan Farnia is partially supported by a grant from the Research Grants Council of the Hong Kong Special Administrative Region, China, Project 14209920, and is partially supported by CUHK Direct Research Grants with CUHK Project No. 4055164 and 4937054. The authors also thank the anonymous reviewers for their helpful feedback and constructive suggestions.

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

# A    Datasheet

Following the framework of [53] and [54], Table 8 provides the additional information about our dataset.

Table 8: Datasheet for our dataset

| Questions | Answers |
|---|---|
| **Motivation** | |
| For what purpose was the dataset created? | To correct uncovered errors and inconsistencies within miniF2F dataset [8] and to further facilitate a challenging benchmark for LLM-based theorem provers and autoformalization models. |
| Who created the dataset and on behalf of which entity? | The authors of this paper. |
| Who funded the creation of the dataset? | The company where the authors work. |
| Any other comment? | None. |
| **Composition** | |
| What do the instances that comprise the dataset represent? | miniF2F theorems, each consisting of 4 components: informal statement and informal proofs in English, formal statements and formal proofs in Lean 4. |
| How many instances are there in total? | 488 theorems |
| Does the dataset contain all possible instances or is it a sample of instances from a larger set? | Yes, we present all instances of the miniF2F dataset. |
| What data does each instance consist of? | A formal theorem in Lean with its formal proof and its informal description and informal proof. |
| Is there a label or target associated with each instance? | Only informal prefix. |
| Is any information missing from individual instances? | No. |
| Are relationships between individual instances made explicit? | Not applicable. |
| Are there recommended data splits? | The dataset follows the same split as the original version, i.e. 244 test instances and 244 validation instances. |
| Are there any errors, sources of noise, or redundancies in the dataset? | No, there are no errors in the proposed dataset to the best of our knowledge. Every formal statement and formal proof compiles with no error in Lean4. All the informal statements and proofs are checked by human. |
| Is the dataset self-contained, or does it link to or otherwise rely on external resources? | The dataset is a self-contained corrected version of miniF2F. |
| Does the dataset contain data that might be considered confidential? | No. |
| Does the dataset contain data that, if viewed directly, might be offensive, insulting, threatening, or might otherwise cause anxiety? | No. |
| **Collection Process** | |
| | Continued on next page |

| | |
|---|---|
| How was the data associated with each instance acquired? | All 488 instances originate from the miniF2F dataset and further augmented to correct mistakes and inconsistencies. Every theorem has a corresponding entry in the source dataset. |
| What mechanisms or procedures were used to collect the data? | The theorems were taken from miniF2F. Original problem statements were taken from official web pages of mathematical competitions such as IMO, AMC, AIME. |
| If the dataset is a sample from a larger set, what was the sampling strategy? | No, there is a one to one match between this dataset and the original miniF2F dataset. |
| Who was involved in the data collection process and how were they compensated? | The dataset was taken from open-source miniF2F dataset. |
| Over what time frame was the data collected? | Dataset was taken directly from miniF2F work. |
| Were any ethical review processes conducted? | No. |
| **Preprocessing/cleaning/labeling** | |
| Was any preprocessing/cleaning/labeling of the data done? | Yes, we performed post processing of the dataset manually to uncover incorrect, misleading, inconsistent or wrong statements. |
| Was the "raw" data saved in addition to the preprocessed/cleaned/labeled data? | The raw data is open-source and available to the public. |
| Is the software that was used to preprocess/clean/label the data available? | We used Lean compiler to type check the formal statements. |
| Any other comments? | No. |
| **Uses** | |
| Has the dataset been used for any tasks already? | We evaluated numerous theorem proving models such as Deepseek-Prover-V1.5-RL, Goedel-Prover-SFT, Kimina-Prover-Distill-7B to evaluate the dataset. Additionally, we performed autoformalization experiments with Herald translator, Kimina autoformalizer and OpenAI o4-mini models. |
| Is there a repository that links to any or all papers or systems that use the dataset? | Public GitHub and HuggingFace links will be provided at a later date. |
| What (other) tasks could the dataset be used for? | The dataset may be used for benchmarking theorem provers and autoformalization models. Other tasks may involve incorporating the dataset into informal-formal theorem proving environments. |
| Is there anything about the composition of the dataset or the way it was collected and preprocessed/cleaned/labeled that might impact future uses? | We do not anticipate future issues emerging from the proposed benchmark dataset. |
| Are there tasks for which the dataset should not be used? | The test set, and preferably validation set, should not be used to train the theorem provers and autoformalizers. |
| Any other comments? | No. |
| **Distribution** | |
| Will the dataset be distributed to third parties outside of the entity on behalf of which the dataset was created? | Yes, we will release the dataset on public platforms such as GitHub and HuggingFace at a later date. |
| How will the dataset will be distributed? | GitHub, HuggingFace. |

| | |
|---|---|
| When will the dataset be distributed? | The dataset will be distributed along with the camera-ready version of the submission. |
| Will the dataset be distributed under a copyright or other intellectual property license, and/or under applicable terms of use? | Yes, the dataset will be released under the MIT license. |
| Have any third parties imposed IP-based or other restrictions on the data associated with the instances? | No. |
| Do any export controls or other regulatory restrictions apply to the dataset or to individual instances? | No. |
| Any other comments? | No. |
| **Maintenance** | |
| Who will be supporting/hosting/maintaining the dataset? | The last author of the submission. |
| How can the owner/curator/manager of the dataset be contacted? | The manager of the dataset may be reached through an email or any other public means, such as GitHub profile. |
| Is there an erratum? | Formal statements do not require erratum, and corrected informal statements are an erratum of the original informal statements sourced from the miniF2F benchmark. |
| Will the dataset be updated? | Yes, we plan to periodically update the dataset as new versions of Lean become available. |
| If the dataset relates to people, are there applicable limits on the retention of the data associated with the instances? | Not applicable. |
| Will older versions of the dataset continue to be supported/hosted/maintained? | Yes. |
| If others want to extend/augment/build on/contribute to the dataset, is there a mechanism for them to do so? | Yes, since we release coupled informal and formal statements, others may expand the dataset to other formal languages, such as Isabelle. Informal statements can also be translated to other languages such as Chinese. The problems may be formalized further to cover more formal theorem proving languages. |
| Any other comments? | No. |

# B  Effect of a clearly structured informal proof as opposed to a vague one

A failure case of Kimina prover is $aime\_1987\_p5$. However, when we provide a more clear informal proof for this theorem, Kimina Prover succeeds in proving it. This indicates the positive effect of a better informal proof on the existing theorem provers, and the higher quality of miniF2F-v2.

---

**aime_1987_p5 with informal proofs**

**miniF2F-v1**

**miniF2F-v2(s/c)**

Find $3x^2y^2$ if $x$ and $y$ are integers such that $y^2 + 3x^2y^2 = 30x^2 + 517$. Show that it is 588.

Find $3x^2y^2$ if $x$ and $y$ are integers such that $y^2 + 3x^2y^2 = 30x^2 + 517$. Show that $3x^2y^2$ is 588.

```
/-Formal Statement-/
theorem aime_1987_p5 (x y : ℤ) (h₀ : y ^ 2 + 3 * (x ^ 2 * y ^ 2) = 30 * x ^
    2 + 517) :
    3 * (x ^ 2 * y ^ 2) = 588 := by
    sorry
```

**Informal proof:**
If we move the $x^2$ term to the left side, it is [[SFFT|factorable]]:
$(3x^2 + 1)(y^2 - 10) = 517 - 10$
507 is equal to $3 \cdot 13^2$. Since $x$ and $y$ are integers, $3x^2 + 1$ cannot equal a multiple of three. 169 doesn't work either, so $3x^2 + 1 = 13$, and $x^2 = 4$. This leaves $y^2 - 10 = 39$, so $y^2 = 49$. Thus, $3x^2y^2 = 3 \times 4 \times 49 = 588$.

**Informal proof:**
From the equation $y^2 + 3x^2y^2 = 30x^2 + 517$ we first rewrite it as

$$3x^2y^2 = 30x^2 + 517 - y^2.$$

Since squares are nonnegative this forces $y^2 \leq 517$, hence $-22 \leq y \leq 22$. There are only finitely many integer choices for $y$ in this range, so we check each one: for each fixed $y$, the rewritten equation becomes a concrete quadratic in $x$ which can be checked by direct computation to yield $3x^2y^2 = 588$. Thus in all cases the desired conclusion holds.

---

## C Wrong formalization because of unfamiliarity with the mathematical definitions in Mathlib

Another failure case of Kimina Prover is $algebra\_cubrtrp1oncubrtreq3\_rcubp1onrcubeq5778$ ($algebra\_5778$ in short). The informal statement for this theorem relies upon a simple definition of cube root common in precollege math. However, in Mathlib, the $n^{th}$ root, via the *rpow* function, is defined using a more advanced definition that is compatible with a broader network of definitions including the real roots of negative complex numbers and the continuity of roots of negative real numbers. The definition of $n^{th}$ root in Mathlib does not correspond to the common definition in precollege math, and therefore, the formalization of this problem in miniF2F is wrong and unprovable. In other words, the *rpow* function in Mathlib is not equivalent to the definition of cube root as stated in the informal statement for this theorem and the use of *rpow* in this formalization makes this theorem unprovable in Lean.

In fact, we write a formal proof giving a counterexample for the case when the variable is negative, proving that this theorem is unprovable in Lean as it appears in miniF2F-v1, i.e., a formal proof for unprovability of this theorem.

As alternative, in the below diagram, we show three correct formalizations of this problem. The first version still relies on the definition of $n^{th}$ root in Lean, but makes the variable nonnegative avoiding any conflict with lemma: $Real.rpow\_def\_of\_neg$ in Mathlib. The second version defines a new $n^{th}$ root function corresponding to precollege math. The third formalization excludes the only possible negative root of equation $h_0$. After correcting the formal statement, Kimina Prover successfully proves this theorem. The proofs of the correct formalization and the proof of counterexample for the incorrect statement in miniF2F wil be released with the paper.

---

Informal Statement: Let $r$ be a real number such that $r^{\frac{1}{3}} + \frac{1}{r^{\frac{1}{3}}} = 3$. Show that $r^3 + \frac{1}{r^3} = 5778$.

**Incorrect Formalization [miniF2F-v1]**
theorem algebra5778
$(r : \mathbb{R})$
$(h_0 : r^{(1/3:\mathbb{R})} + 1/r^{(1/3:\mathbb{R})} = 3 :$
$r^3 + 1/r^3 = 5778 :=$ by

**Correct Formalization #1**
theorem algebra5778
$(r : \mathbb{R})$
$(h_0 : r^{(1/3:\mathbb{R})} + 1/r^{(1/3:\mathbb{R})} = 3)$
$(h_1 : 0 \le r) :$
$r^3 + 1/r^3 = 5778 :=$ by

**Correct Formalization #2 [miniF2F-v2(s/c)]**
theorem algebra5778
$(r : \mathbb{R})$ $(qpow : \mathbb{R} \to \mathbb{Q} \to \mathbb{R})$
$(hq : qpow = fun\ x\ q \mapsto if\ 0 \le x\ then$
$x.rpow(\uparrow a)\ else\ -(-x).rpow(\uparrow a))$
$(h_0 : qpow\ r\ (1/3) + 1/qpow\ r\ (1/3) = 3) :$
$r^3 + 1/r^3 = 5778 :=$ by

**Correct Formalization #3**
theorem algebra5778
$(r : \mathbb{R})$
$(h_0 : r^{(1/3:\mathbb{R})} + 1/r^{(1/3:\mathbb{R})} = 3)$
$(h_1 : r^{(1/3:\mathbb{R})} \ne (1/2)(-r)^{(1/3:\mathbb{R})}) :$
$r^3 + 1/r^3 = 5778 :=$ by

---

## D Examples of autoformalizer outputs on miniF2F v1 and v2

In this section, we present three examples of modified problems and analyze their impact on autoformalization models. We show that supplying corrected miniF2F informal statements can significantly affect model performance, and that the specific nature of each formalization error drives different model behaviors.

### D.1 `mathd_algebra_31`

The original informal statement of `mathd_algebra_31` omits several critical details and is defined ambiguously. In particular, it denotes a limit by "...", leaving the definition ambiguous. When tasked with this problem, both Herald and Kimina attempt to encode the underlying recursive function explicitly, and fail.

To fix these issues, we provide a clearly specified version of the problem with an explicit recursive definition. After this revision, Kimina successfully produces a correct autoformalization although Herald still fails. These results demonstrate that clearly written (i.e., higher quality) informal statements improves the reliability of autoformalization benchmarks, and it can improve the accuracy of existing models, too.

### D.2 `imo_1960_p2`

The original informal statement does not specify the solution to the question, i.e., the proof goal. In other words, the question asks the the examinees to first find the solution and then prove its correctness. With such an informal statement, autoformalization models should also leave the goal undefined using an additional sorry within the formal statement. This is what we do in our miniF2F-v2c. In miniF2F-v2s, however, we amend the informal statement with the answer so that it matches the formal statement. Clearly, the formal and informal statements in miniF2F-v2c are more difficult both for translation and for proving.

At the same time, with our modification of informal statement in miniF2F-v2s for this problem, both Herald and Kimina successfully produce correct formalizations. Although better informal-formal match increases the complexity for automated provers, our results demonstrate that faithful, detailed translations improve some autoformalization attempts.

Interestingly, since o4-mini is an advanced reasoning model, it automatically finds the solution for the statement in miniF2F-v2c and incorporates the answer as the goal in its formal translation. This additional step that o4-mini takes, however, simplifies the problem along the way of translation and makes its formal proof much easier. This may or may not be desirable in various contexts. As explained earlier in the paper, simplifying a problem during translation does not receive credit in our evaluation setting.

### D.3 `amc12b_2003_p6`

In this example, the informal statement is correct. However, the formal statement is not a correct translation of it. The formal statement asks for an extra solution, and drops the intricate detail of "a possible solution" in the informal statement. This can be considered a mismatch between the informal and formal statements. Despite the fact that each of them is correct and provable, the formal and informal statements in the original miniF2F do not translate to each other. One possible approach is to keep the correct informal statement and change the formal statement to correspond to it which we do in miniF2F-v2c. In miniF2F-v2s, however, we chose to change the formal statement so that it corresponds to the formal one. After this update, both Herald and o4-mini correctly autoformalize the problem in miniF2F-v2s. For miniF2F-v2c, we changed the formal statement to reflect the multiple-choice nature of the question. All options are equivalent except for the numerical solutions they correspond to. To preserve space, we replaced repeating blocks with "...". Only o4-mini was able to correctly autoformalize given statement.

**miniF2F-v1**

**miniF2F-v2(s/c)**

If $\sqrt{x + \sqrt{x + \sqrt{x + \sqrt{x + \cdots}}}} = 9$, find $x$. Show that it is 72.

Consider a nonnegative real number x, and a recursive function u from natural numbers to real numbers. For all natural numbers, we have $u(n+1) = \sqrt{x + u(n)}$. Assume that the as n reaches infinity in its limit, u(n) goes to 9. What is the value of x? Show that it is 72.

```
/-Formal statement-/
theorem mathd_algebra_31_minif2f_v1
    (x : NNReal)
    (u : ℕ → NNReal)
    (h₀ : ∀ n, u (n + 1) =
    NNReal.sqrt (x + u n))
    (h₁ : Filter.Tendsto u
    Filter.atTop (nhds 9)) :
    9 = NNReal.sqrt (x + 9) := by
    sorry
```

```
/-Formal statement-/
theorem mathd_algebra_31_minif2f_v2
    (x : ℝ) (hx : 0 ≤ x)
    (u : ℕ → ℝ)
    (h₀ : ∀ n, u (n + 1) = Real.sqrt
    (x + u n))
    (h₁ : Filter.Tendsto u
    Filter.atTop (nhds 9)) :
    x = 72 := by
    sorry
```

```
/-Kimina Formalized - miniF2F-v1-/
theorem mathd_algebra_31_kimina
    {x : ℝ}
    (hx : 0 ≤ x)
    (h : sqrt(x + sqrt(x + sqrt(x +
    sqrt(x + sqrt(x + 0))))) = 9) :
    x = 72 := by
    sorry
```

```
/-Kimina Formalized - miniF2F-v2s-/
theorem mathd_algebra_31_kimina
    (x : ℝ) (hx : 0 ≤ x) (u : ℕ → ℝ)
    (hu : ∀ n, u (n + 1) = sqrt (x +
    u n))
    (h : Tendsto u atTop (nhds 9)) :
    x = 72 := by
    sorry
```

```
/-Herald Formalized - miniF2F-v1-/
theorem mathd_algebra_31_herald :
    sqrt(x + sqrt(x + sqrt(x +
    sqrt(x + 9)))) = 9 ↔ x = 72   :=
      by
  sorry
```

```
/-Herald Formalized - miniF2F-v2s-/
theorem mathd_algebra_31_herald (x :
    ℝ) (h : 0 < x) (hx : Σ' i : ℕ,
    (Real.sqrt (x + (Real.sqrt (x +
    (Real.sqrt (x + (Real.sqrt (x +
    0)))))))) = 9) : x = 72 := by
    sorry
```

```
/-o4-mini Formalized - miniF2F-v1-/
theorem mathd_algebra_31_o4mini (x y
    : ℝ) (h1 : y = Real.sqrt (x +
    y)) (h2 : y = 9) : x = 72 := by
    sorry
```

```
/-o4-mini Formalized - miniF2F-v2s-/
theorem mathd_algebra_31_o4mini {x :
    ℝ} (h : 9 = Real.sqrt (x + 9)) :
    x = 72 := by
    sorry
```

**miniF2F-v1**

**miniF2F-v2s**

For what values of the variable $x$ does the following inequality hold:
$$\frac{4x^2}{(1 - \sqrt{2x+1})^2} < 2x + 9 \text{ ?}$$

Let $x$ be a real number. Assume that:
- $1 + 2x \geq 0$ (so that the square root is defined), - $(1 - \sqrt{1+2x})^2 \neq 0$ (so the denominator is nonzero), and - $\frac{4x^2}{(1-\sqrt{1+2x})^2} < 2x + 9$.
Prove that:
$$-\frac{1}{2} \leq x \quad \text{and} \quad x < \frac{45}{8} .$$

```
/-Formal Statement-/
theorem imo_1960_p2 (x : ℝ) (h₀ : 0 ≤ 1 + 2 * x) (h₁ : (1 - Real.sqrt (1 +
    2 * x)) ^ 2 ≠ 0)
    (h₂ : 4 * x ^ 2 / (1 - Real.sqrt (1 + 2 * x)) ^ 2 < 2 * x + 9) : -(1 /
    2) ≤ x ∧ x < 45 / 8 := by
  sorry
```

```
/-Kimina Formalized - miniF2F-v1-/
theorem imo_1960_p2_kimina (x : ℝ)
    (hx : 0 ≤ 2 * x + 1 ∧ 1 - sqrt
    (2 * x + 1) ≠ 0) :
  (4 * x ^ 2 / (1 - sqrt (2 * x +
    1)) ^ 2 < 2 * x + 9 ↔ x < 1 / 2
    ∨ x > 7 / 2) := by
  sorry
```

```
/-Kimina Formalized - miniF2F-v2s-/
theorem imo_1960_p2_kimina (x : ℝ)
    (h₀ : 1 + 2 * x ≥ 0) (h₁ : (1 -
    Real.sqrt (1 + 2 * x))^2 ≠ 0)
    (h₂ : 4 * x^2 / (1 - Real.sqrt
    (1 + 2 * x))^2 < 2 * x + 9) :
  -1 / 2 ≤ x ∧ x < 45 / 8 := by
  sorry
```

```
/-Herald Formalized - miniF2F-v1-/
theorem imo_1960_p2_herald (x : ℝ) :
    (4 * x ^ 2 / (1 - Real.sqrt (2 *
    x + 1)) ^ 2 < 2 * x + 9) ↔ x <
    11 / 2    := by
  sorry
```

```
/-Herald Formalized - miniF2F-v2s-/
theorem imo_1960_p2_herald (x : ℝ)
    (h₁ : 1 + 2 * x ≥ 0) (h₂ : (1 -
    Real.sqrt (1 + 2 * x)) ^ 2 ≠ 0)
    (h₃ : 4 * x ^ 2 / (1 - Real.sqrt
    (1 + 2 * x)) ^ 2 < 2 * x + 9) :
    -1 / 2 ≤ x ∧ x < 45 / 8    := by
  sorry
```

```
/-o4-mini Formalized - miniF2F-v1-/
theorem imo_1960_p2_o4mini (x : ℝ)
    (h_nonneg : 2 * x + 1 ≥ 0)
    (h_ne_zero : 1 - Real.sqrt (2 *
    x + 1) ≠ 0) :
  4 * x ^ 2 / (1 - Real.sqrt (2 * x +
    1)) ^ 2 < 2 * x + 9 ↔
  -1 / 2 ≤ x ∧ x < 45 / 8 ∧ x ≠ 0 :=
    by
  sorry
```

```
/-o4-mini Formalized - miniF2F-v2s-/
theorem imo_1960_p2 (x : ℝ)
    (h1 : 1 + 2 * x ≥ 0)
    (h2 : (1 - Real.sqrt (1 + 2 * x))
    ^ 2 ≠ 0)
    (h3 : 4 * x ^ 2 / (1 - Real.sqrt
    (1 + 2 * x)) ^ 2 < 2 * x + 9) :
    -1 / 2 ≤ x ∧ x < 45 / 8 := by
  sorry
```

## miniF2F-v1

The second and fourth terms of a geometric sequence are 2 and 6. Which of the following is a possible first term?

**(A)** $-\sqrt{3}$   **(B)** $-\dfrac{2\sqrt{3}}{3}$

**(C)** $-\dfrac{\sqrt{3}}{3}$   **(D)** $\sqrt{3}$   **(E)** 3

Show that it is **(B)** $-\dfrac{2\sqrt{3}}{3}$.

## miniF2F-v2s

The second and fourth terms of a geometric sequence are 2 and 6. Show that the first term is either $-\dfrac{2\sqrt{3}}{3}$ or $\dfrac{2\sqrt{3}}{3}$.

## miniF2F-v2c

The second and fourth terms of a geometric sequence are 2 and 6. Which of the following is a possible first term? Prove that it is one of the following options.

**(A)** $-\sqrt{3}$   **(B)** $-\dfrac{2\sqrt{3}}{3}$

**(C)** $-\dfrac{\sqrt{3}}{3}$   **(D)** $\sqrt{3}$   **(E)** 3

---

/–Formal Statement–/
```
theorem amc12b_2003_p6_v1 (a r : ℝ) (u :
      ℕ → ℝ)
  (h₀ : ∀ k, u k = a * r ^ k) (h₁ : u 1 = 2)
  (h₂ : u 3 = 6) : u 0 = 2 / Real.sqrt 3 ∨ u 0 =
      -(2 / Real.sqrt 3) := by
  sorry
```

/–Formal Statement–/
```
theorem amc12b_2003_p6_v2s (a r : ℝ) (u :
      ℕ → ℝ)
  (h₀ : ∀ k, u k = a * r ^ k) (h₁ : u 1 = 2)
  (h₂ : u 3 = 6) : u 0 = 2 / Real.sqrt 3 ∨ u 0 =
      -(2 / Real.sqrt 3) := by
  sorry
```

/–Formal Statement–/
```
theorem amc12b_2003_p6_v2c :
  (∃ (u : ℕ → ℝ) (a : ℝ) (r : ℝ), ∀ k, u k = a *
      r ^ k ∧ u 1 = 2 ∧ u 3 = 6 ∧ u 0 = -
      Real.sqrt 3) ∨
  (∃ . . . = - 2 * Real.sqrt 3 / 3) ∨
  (∃ . . . = - Real.sqrt 3 / 3) ∨
  (∃ . . . = Real.sqrt 3) := by
  sorry
```

---

/–Kimina Formalized – miniF2F–v1–/
```
theorem amc12b_2003_p6_kimina (a : ℕ
      → ℝ) (h : ∃ r, ∀ n, a n = a 0 * r ^ n)
  (ha : a 1 = 2) (hb : a 3 = 6) :
  a 0 = -2 * sqrt 3 / 3 := by
  sorry
```

/–Kimina Formalized – miniF2F–v2s–/
```
theorem amc12b_2003_p6_kimina (a r : ℝ)
  (h₀ : r ≠ 0) (h₁ : a * r = 2) (h₂ : a *
      r^3 = 6) :
  a = 2 * Real.sqrt 3 / 3 ∨ a = -2 *
      Real.sqrt 3 / 3 := by
  sorry
```

/–Kimina Formalized – miniF2F–v2c–/
```
theorem amc12b_2003_p6_kimina (a q : ℝ)
  (hq : q ≠ 0) (ha : a ≠ 0) (h1 : a*q =
      2) (h2 : a*q^3 = 6) : a = -sqrt 3 ∨ a
      = -2*sqrt 3 ∨ a = -sqrt 3 / 3 ∨
  a = sqrt 3 ∨ a = 3 := by
  sorry
```

---

/–Herald Formalized – miniF2F–v1–/
```
theorem amc12b_2003_p6_herald (a : ℝ)
  (h₀ : a * r = 2)
  (h₁ : a * r^3 = 6) :
  r = 3 ∧ a = -2 * Real.sqrt 3 / 3 := by
  sorry
```

/–Herald Formalized – miniF2F–v2s–/
```
theorem amc12b_2003_p6_herald (a : ℕ →
      ℝ)
  (h : a 2 = 2 ∧ a 4 = 6) : a 1 = -2 *
      Real.sqrt 3 / 3 ∨ a 1 = 2 *
      Real.sqrt 3 / 3 := by
  sorry
```

/–Herald Formalized – miniF2F–v2c–/
```
theorem amc12b_2003_p6 (a : ℝ)
  (h₀ : a * r = 2)
  (h₁ : a * r^3 = 6) :
  a = -Real.sqrt 3 ∨ a = -2 * Real.sqrt 3 /
      3 ∨ a = -Real.sqrt 3 / 3 ∨ a =
      Real.sqrt 3 ∨ a = 3 := by
  sorry
```

---

/–o4–mini Formalized – miniF2F–v1–/
```
theorem amc12b_2003_p6_o4mini : ∃ (a r
      : ℝ), r ≠ 0 ∧ a * r = 2 ∧ a * r ^ 3 =
      6 ∧ a = -2 * sqrt 3 / 3 := by
  sorry
```

/–o4–mini Formalized – miniF2F–v2s–/
```
theorem amc12b_2003_p6_o4mini (a r : ℝ)
  (h1 : a * r = 2) (h2 : a * r^3 = 6) :
  a = 2 * Real.sqrt 3 / 3 ∨ a = -2 *
      Real.sqrt 3 / 3 := by
  sorry
```

/–o4–mini Formalized – miniF2F–v2c–/
```
theorem amc12b_2003_p6_o4mini :
  ∃ (a r : ℝ),
  a * r = 2 ∧
  a * r^3 = 6 ∧
  (a = -Real.sqrt 3 ∨ a = -2 * Real.sqrt 3
      / 3 ∨ a = -Real.sqrt 3 / 3 ∨ a =
      Real.sqrt 3 ∨ a = 3) := by
  sorry
```

# E  Examples of modified statements

## E.1  `induction_pord1p1on2powklt5on2`

The original problem was unprovable due to a missing pair of parentheses, which led to an incorrect formalization. We corrected this error in miniF2F-v2 by inserting the appropriate parentheses.

## E.2  `aime_1990_p4`

In the original formal statement, three additional hypotheses $(h_1, h_2, h_3)$ explicitly assert that certain expressions are nonzero, which simplifies proof generation for theorem provers. These are extra assumptions that are not present in the informal statement that was given to the participants of AIME 1990, and they are not necessary to prove the theorem. To remove this discrepancy between the formal and informal statements, we remove the added hypothesis from the formal statement. This makes the theorem more challenging for theorem provers as they have to prove each of those hypothesis as intermediate steps to prove the theorem.

---

[Unprovable] Comparison of induction_pord1p1on2powklt5on2 across miniF2F-v1 and miniF2Fv2

Show that for positive integer $n$, $(\prod_{k=1}^{n}(1+1/2^k)) < 5/2$.

**miniF2F-v1**     **miniF2F-v2(s/c)**

```
/-Formal Statement - miniF2F-v1-/
theorem
    induction_pord1p1on2powklt5on2
  (n : ℕ) (h₀ : 0 < n) :
  (∏ k in Finset.Icc 1 n, 1 + (1 : ℝ
    ) / 2 ^ k) < 5 / 2 := by
  sorry
```

```
/-Formal Statement - miniF2F-v2-/
theorem
    induction_pord1p1on2powklt5on2
  (n : ℕ) (h₀ : 0 < n) :
  (∏ k ∈ Finset.Icc (1:ℕ) n, ((1 :
    ℝ) + (1 : ℝ) / 2 ^ k)) < (5 / 2
    : ℝ) := by
  sorry
```

---

[Simplified] Comparison of aime_1990_p4 across miniF2F-v1 and miniF2Fv2

Find the positive solution to $\frac{1}{x^2-10x-29} + \frac{1}{x^2-10x-45} - \frac{2}{x^2-10x-69} = 0$. Show that it is 13.

**miniF2F-v1**     **miniF2F-v2(s/c)**

```
/-Formal Statement - miniF2F-v1-/
theorem aime_1990_p4
  (x : ℝ) (h₀ : 0 < x)
  (h₁ : x ^ 2 - 10 * x - 29 ≠ 0)
  (h₂ : x ^ 2 - 10 * x - 45 ≠ 0)
  (h₃ : x ^ 2 - 10 * x - 69 ≠ 0)
  (h₄ : 1 / (x ^ 2 - 10 * x - 29) +
    1 / (x ^ 2 - 10 * x - 45) - 2 /
    (x ^ 2 - 10 * x - 69) = 0) :
  x = 13 := by
  sorry
```

```
/-Formal Statement - miniF2F-v2-/
theorem aime_1990_p4
  (x : ℝ) (h₀ : 0 < x)
  (h₄ : 1 / (x ^ 2 - 10 * x - 29) +
    1 / (x ^ 2 - 10 * x - 45) - 2 /
    (x ^ 2 - 10 * x - 69) = 0) :
  x = 13 := by
  sorry
```

---

## E.3  `amc12b_2021_p9`

In this example, we retained each logarithm in its base form, writing $\log_c a$ directly rather than converting it to $\frac{\log a}{\log c}$, so that the formal statement exactly corresponds to the informal one as it

appeared in the AMC. These changes yield a slightly more challenging version of the problem. And we observe that all three theorem proving models, Deepseek-Prover-V1.5-RL, Goedel-Prover-SFT and Kimina-Prover-Preview-Distill-7B, fail on the modified version of this theorem.

### E.4 `mathd_algebra_487`

In the original miniF2F version, the problem was oversimplified: it introduced four variables (instead of two) and omitted the Euclidean-space context, therefore providing extra hints that eases proof generation task for LLMs. In miniF2F-v2, we restore the two-variable formulation over $\mathbb{R}^2$, remove these implicit assumptions. This leads to a spike in the problem's difficulty. These more faithful and challenging instances yield a more rigorous benchmark for evaluating theorem provers.

## [Simplified] Comparison of amc12b_2021_p9 across miniF2F-v1 and miniF2Fv2

**miniF2F-v1**

What is the value of $\frac{\log_2 80}{\log_{40} 2} - \frac{\log_2 160}{\log_{20} 2}$?

$(\textbf{A})0 \qquad (\textbf{B})1 \qquad (\textbf{C})\frac{5}{4}$

$(\textbf{D})2 \qquad (\textbf{E})\log_2 5$

Show that it is (D).

```
/-Formal Statement -
    miniF2F-v1-/
theorem amc12b_2021_p9 :
  Real.log 80 /
    Real.log 2 /
    (Real.log 2 /
    Real.log 40) -
  Real.log 160 /
    Real.log 2 /
    (Real.log 2 /
    Real.log 20)
  = 2 := by
  sorry
```

**miniF2F-v2s**

What is the value of $\frac{\log_2 80}{\log_{40} 2} - \frac{\log_2 160}{\log_{20} 2}$?
Show that it is 2.

```
/-Formal Statement -
    miniF2F-v2s-/
theorem amc12b_2021_p9 :
  Real.logb 2 80 /
    (Real.logb 40 2) -
    Real.logb 2 160 /
    Real.logb 20 2) = 2
    := by
  sorry
```

**miniF2F-v2c**

What is the value of $\frac{\log_2 80}{\log_{40} 2} - \frac{\log_2 160}{\log_{20} 2}$?
Prove that it is one of the following options.

$(\textbf{A})0 \qquad (\textbf{B})1 \qquad (\textbf{C})\frac{5}{4}$

$(\textbf{D})2 \qquad (\textbf{E})\log_2 5$

```
/-Formal Statement -
    miniF2F-v2c-/
theorem amc12b_2021_p9
  (X : ℝ)
  (hX : X = Real.logb 2
    80 / (Real.logb 40
    2) - Real.logb 2
    160 / Real.logb 20
    2) :
  X = 2 ∨ X = 4 ∨ X =
    6 ∨ X = 30 ∨ X =
    32 := by
  sorry
```

## [Excessively Simplified] Comparison of mathd_algebra_487 across miniF2F-v1 and miniF2Fv2

What is the distance between the two intersections of $y = x^2$ and $x + y = 1$? Show that it is $\sqrt{10}$.

**miniF2F-v1**

```
/-Formal Statement - miniF2F-v1-/
theorem mathd_algebra_487
  (a b c d : ℝ) (h₀ : b = a ^ 2)
  (h₁ : a + b = 1) (h₂ : d = c ^ 2)
  (h₃ : c + d = 1) (h₄ : a ≠ c) :
  Real.sqrt ((a - c) ^ 2 + (b - d) ^
    2) = Real.sqrt 10 := by
  sorry
```

**miniF2F-v2(s/c)**

```
/-Formal Statement - miniF2F-v2-/
theorem mathd_algebra_487
  (F G I : Set (EuclideanSpace ℝ
    (Fin 2)))
  (hF : F = { x | x 1 = (x 0) ^ 2})
  (hG : G = { x | x 0 + x 1 = 1})
  (hI : I = (F ∩ G))
  (A B : EuclideanSpace ℝ (Fin 2))
  (h₀ : ∀ x, x ∈ I ↔ x = A ∨ x = B):
  dist A B = Real.sqrt 10 := by
  sorry
```

# F Examples of challenging miniF2F-v2c problems

## F.1 `imo_1983_p6`

In this problem, we did not modify the informal statement; instead, we corrected the paired formal statement. The original task is to prove an inequality and determine when equality occurs. In v1, the formal statement reflects only the first part, omitting the statement and the proof of when the equality holds. We modified the given hypotheses to match the problem formulation, but one could argue that the original formulation is also correct. More importantly, we ask the model to perform two tasks: prove the inequality and determine the values of $a$, $b$, and $c$ to achieve equality. Now, when asked to perform both tasks, the current medium-sized state-of-the-art prover, Deepseek-Prover-V2-7B, fails and no longer produces a correct proof. We believe that the introduced changes are more faithful to the intended difficulty and reflect the limitations of current ATP models.

## F.2 `imo_1997_p5`

Another type of change made exclusively in miniF2F-v2c is the correction of both informal and formal statements to match the intended difficulty. In v1, the imo_1997_p5 problem statement is simplified and reworded to match the formal statement. Moreover, instead of requiring the prover to find the solution and prove it, the formal statement provides a clear goal. We changed both the informal and formal statements to reflect the original imo_1997_p5 formulation and tasked the models with coming up with the correct goal and writing a valid proof. Our results suggest that when the goal is unknown, the models struggle to find valid proofs, since they essentially must solve two separate tasks to achieve the result.

---

**[Removed solution from formal statement] Comparison of imo_1983_p6 across miniF2F-v1 and miniF2F-v2c**

Let $a$, $b$ and $c$ be the lengths of the sides of a triangle. Prove that:

$$a^2 b(a - b) + b^2 c(b - c) + c^2 a(c - a) \geq 0.$$

Determine when equality occurs.

**miniF2F-v1**

**miniF2F-v2c**

```
/-Formal Statement - miniF2F-v1-/
theorem imo_1983_p6
(a b c : ℝ) (h₀ : 0 < a ∧ 0 < b ∧ 0
    < c)
(h₁ : c < a + b) (h₂ : b < a + c)
(h₃ : a < b + c) : 0 ≤ a ^ 2 * b *
    (a - b) + b ^ 2 * c * (b - c) +
    c ^ 2 * a * (c - a) := by
  sorry
```

```
/-Formal Statement - miniF2F-v2-/
abbrev imo_1983_p6_solution : ℝ → ℝ
    → ℝ → Prop := sorry

theorem imo_1983_p6
(T : Affine.Triangle ℝ
    (EuclideanSpace ℝ (Fin 2))) :
let a := dist (T.points 1) (T.points
    2)
let b := dist (T.points 0) (T.points
    2)
let c := dist (T.points 0) (T.points
    1)
0 ≤ a^2 * b * (a - b) + b^2 * c * (b
    - c) + c^2 * a * (c - a) ∧
(0 = a^2 * b * (a - b) + b^2 * c *
    (b - c) + c^2 * a * (c - a) ↔
    imo_1983_p6_solution a b c) := by
  sorry
```

[Simplified informal and formal statements] Comparison of imo_1997_p5 across miniF2F-v1 and miniF2F-v2c

**miniF2F-v1**

Show that if $x$ and $y$ are positive integers such that $x^{y^2} = y^x$, then $(x, y)$ is equal to $(1, 1)$, $(16, 2)$, or $(27, 3)$.

```
/-Formal Statement - miniF2F-v1-/
theorem imo_1997_p5
(x y : ℕ) (h₀ : 0 < x ∧ 0 < y) (h₁ :
    x ^ y ^ 2 = y ^ x) :
(x, y) = (1, 1) ∨ (x, y) = (16, 2) ∨
    (x, y) = (27, 3) := by
  sorry
```

**miniF2F-v2c**

Find all pairs $(a, b)$ of integers $a, b \geq 1$ that satisfy the equation $x^{y^2} = y^x$.

```
/-Formal Statement - miniF2F-v2-/
abbrev imo_1997_p5_solution : Set
    (ℕ×ℕ) := sorry

theorem imo_1997_p5
(x y : ℕ)
(h₀ : 1 ≤ x ∧ 1 ≤ y) :
x ^ y ^ 2 = y ^ x ↔ (x, y) ∈
    imo_1997_p5_solution := by
  sorry
```

### F.3  amc12_2000_p12

In our efforts to make the problems closer to Math Olympiad settings, we removed the correct answers from MCQ-style questions and reformulated the goals to require first selecting the correct solution and then proving it.

[Omitted correct answers from multiple choice problems] Comparison of amc12_2000_p12 across miniF2F-v1 and miniF2F-v2c

**miniF2F-v1**

Let $A, M,$ and $C$ be nonnegative integers such that $A+M+C = 12$. What is the maximum value of $A \cdot M \cdot C + A \cdot M + M \cdot C + A \cdot C$?

(A) 62 (B) 72 (C) 92 (D) 102 (E) 112

Show that it is E.

```
/-Formal Statement - miniF2F-v1-/
theorem amc12_2000_p12
(a m c : ℕ) (h₀ : a + m + c = 12) :
a * m * c + a * m + m * c + a * c ≤
    112 := by
  sorry
```

**miniF2F-v2c**

Let $A, M,$ and $C$ be nonnegative integers such that $A+M+C = 12$. What is the maximum value of $A \cdot M \cdot C + A \cdot M + M \cdot C + A \cdot C$? Prove that it is one of the following options.

(A) 62 (B) 72 (C) 92 (D) 102 (E) 112

```
/-Formal Statement - miniF2F-v2-/
theorem amc12_2000_p12
(S: Set ℕ)
(hS: S = {x | ∃ a m c : ℕ, (x = a *
    m * c + a * m + m * c + a * c) ∧
    a + m + c = 12}) :
IsGreatest S 62 ∨ IsGreatest S 72 ∨
    IsGreatest S 92 ∨ IsGreatest S
    102 ∨ IsGreatest S 112 := by
  sorry
```

# G  Effect of introduced changes to ATP performance on Olympiad style questions

To investigate the scale of difficulty, we showcase how the theorem provers perform specifically on IMO, AMC and AIME problems. Figure 3 compared four theorem provers across three miniF2F versions. We note that the strongest provers, Deepseek-Prover-V2-7B and Goedel-V2-7B, show a substantial decline in performance solving twice as less IMO problems and 15-50% accuracy decline on AMC. We note that many corrected problems in miniF2F-v2c come from IMO and AMC but not AIME, therefore the performance drop in AIME is small across all theorem provers.

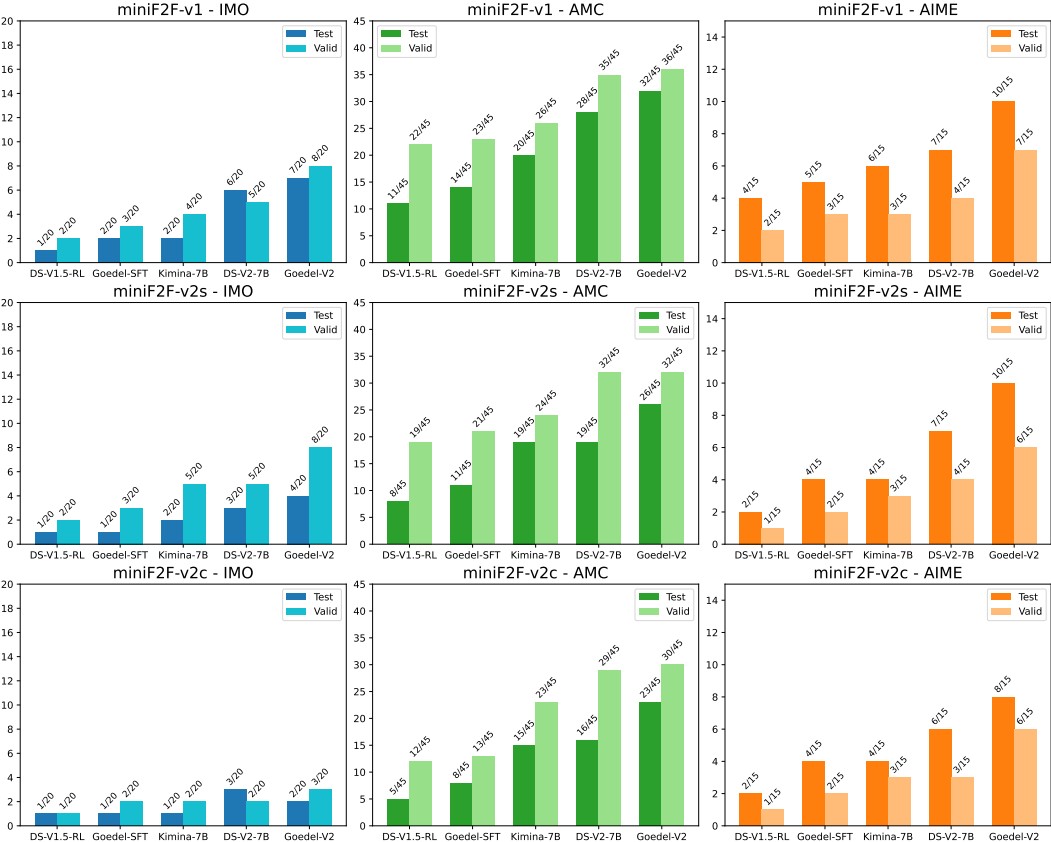

Figure 3: Distribution of solved Olympiad-level (IMO, AMC, AIME) competition problems present in miniF2F-v1/2s/2c across four theorem provers. Theorem prover names were shortened from their original names. Each bar plot also shows the total number of problems present in the dataset.

# H Statistics about our modifications

We present the distribution of uncovered errors and inconsistencies in Figure 4. We observe that the majority of the problems in both test and validation sets are simplified and do not reflect the intended difficulty of the problems. Moreover, approximately 40% of formal statements across both sets contained an error, making the evaluation of LLMs on this benchmark less reliable.

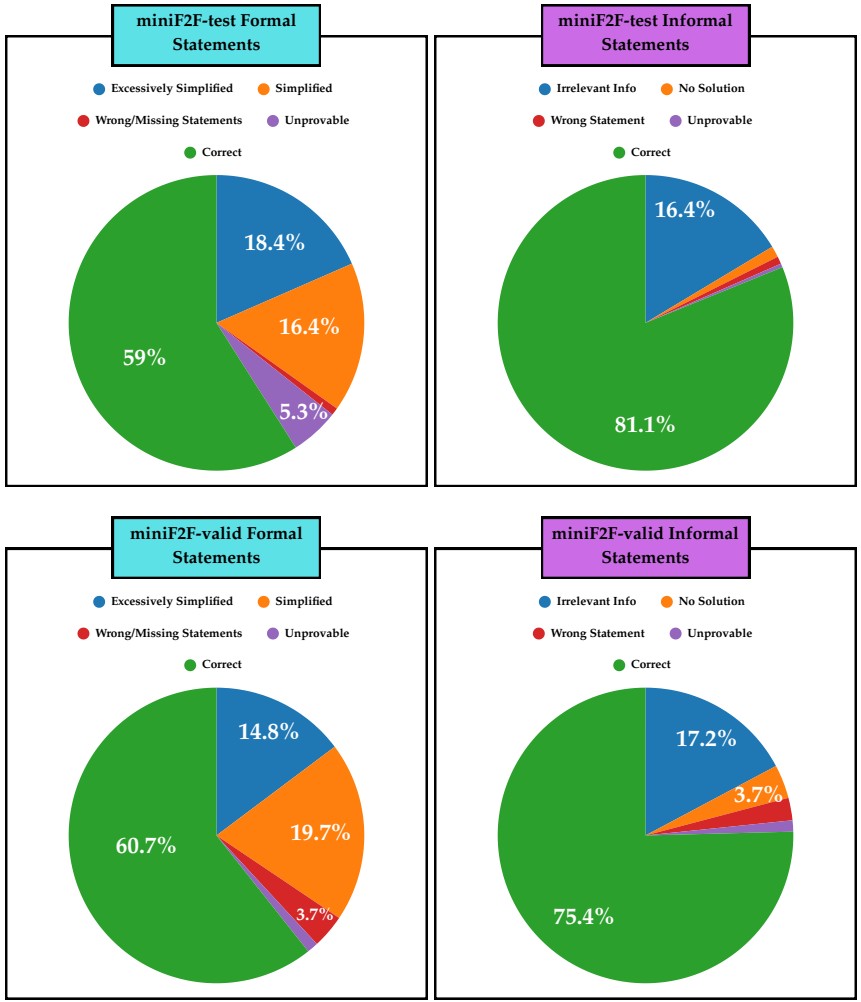

Figure 4: Pie charts of identified formal and informal statement errors within miniF2F benchmark across test and validation sets.

# I   Failure topics of tested autoformalization models

To gain deeper insight into the formalization errors of the tested autoformalizer models, we present our classifications in Tables 9, 10, and 11. The categorization of failure cases was performed manually by Lean experts, except for the "Not validated by Lean4 REPL" category, which corresponds to translations that failed compiler verification.

Table 9: Frequency of HERALD autoformalization errors the miniF2F-test dataset.

| Failure case classification | Frequency |
| --- | --- |
| Wrong/missing/incomplete statements or translations | 17.5% |
| Type wrong/mismatch | 16.25% |
| Max/Min of sets/functions wrongly formalized | 16.25% |
| Incomplete set of assumptions | 12.5% |
| Not validated by Lean4 REPL | 7.5% |
| Digits | 7.5% |
| Finite sets | 3.75% |
| Additional assumptions/hypothesis | 3.75% |
| Common divisors | 2.5% |
| Sequences | 2.5% |
| Others | 10.0% |

Table 10: Frequency of Kimina-autoformalizer autoformalization errors the miniF2F-test dataset.

| Failure case classification | Frequency |
| --- | --- |
| Wrong/missing/incomplete statements or translations | 57.14% |
| Not validated by Lean4 REPL | 28.57% |
| Max/Min of sets/functions wrongly formalized | 7.14% |
| Additional assumptions/hypothesis | 7.14% |

Table 11: Frequency of o4-mini autoformalization errors the miniF2F-test dataset.

| Failure case classification | Frequency |
| --- | --- |
| Not validated by Lean4 REPL | 80.43% |
| Wrong/missing/incomplete statements or translations | 6.52% |
| Max/Min of sets/functions wrongly formalized | 6.52% |
| Incomplete set of assumptions | 2.17% |
| Euclidean spaces | 2.17% |
| Modulus | 1.09% |
| Digits | 1.09% |

## J Toward improving autoformalization models

Improving the performance of autoformalization models requires progress beyond benchmark evaluation. A crucial first step is to adopt rigorous and transparent evaluation practices using high-quality, discrepancy-free benchmarks. This paper takes that step by providing a more reliable basis for assessing model accuracy and consistency.

The next important direction is to develop high-quality training data that accurately aligns formal and informal mathematical statements. Existing datasets often contain discrepancies or are partially closed-source, which can introduce noise during training. Because the miniF2F validation set is frequently reused for training, a more consistent benchmark can have a direct positive effect on model development. Any training example that misaligns formal and informal statements may negatively influence model accuracy and generalization.

Beyond data quality, several factors are likely to affect model performance, including the distribution of the training corpus (e.g., mathematical topics and sample diversity), model architecture, reasoning mechanisms, context length, and the use of automated feedback during training. These aspects have been shown to play critical roles in the performance of large language models and general machine learning systems. Incorporating these insights into future autoformalization research can help identify effective design choices.

Finally, the field would benefit from systematic ablation studies that isolate the contribution of individual components such as data quality, architecture, and reasoning modules. Reliable benchmarks are essential for such analyses, as dataset discrepancies can obscure interpretation. We hope that the availability of correct and comprehensive benchmarks will enable more informative and reproducible studies in this emerging research area.

## K Responsibility statement and License information

We plan to release our dataset under the MIT license on GitHub and Hugging Face. The authors of this submission bear the responsibility in case of rights violation.

