# OpenReview forum: "miniF2F-Lean Revisited: Reviewing Limitations and Charting a Path Forward"
_NeurIPS.cc/2025/Conference — NeurIPS 2025 poster_

### Official Review · Reviewer_Vohv · 2025-06-17

**Clarity:** 3
**Significance:** 3
**Originality:** 2
**Rating:** 4
**Confidence:** 3

**Summary:**

The authors evaluate end-to-end formal reasoning pipelines that read mathematical problems in natural language, formalize them in Lean, and generate proofs. They find that state-of-the-art models achieve only 36% accuracy on this complete pipeline, despite individual components reporting much higher accuracies (97% for autoformalization, 69% for theorem proving). Through extensive analysis, they identify significant discrepancies between formal and informal statements in over half of the miniF2F problems. They correct over 300 Lean statements, fix 16 unprovable theorems, and release miniF2F-v2, which improves end-to-end accuracy to 57%.

**Questions:**

- This is a benchmark study, so why don't you submit to the D&B track? Clearly this paper lacks significant theoretical analysis, it might be more fit to the  D&B track instead of the main track.

- Figure 1 is not necessary as it conveys little information and it occupies a large space in the manuscript. I would suggest to remove it so you can write more content.

- What's the difference between "Translation failure" and "Autoformalization failure"? In line 49-50, you defined Autoformalization as a kind of translation.

- Can you provide the distribution of systematic errors in the original miniF2F in the main text instead of appendx? Though mentioned, you did not specifiy the location of this distribution in the appendix, which makes reading really inconvenient.

- The sentence spaning line 55-59 is too long, which makes it really difficult to undertand. Please break a long sentence into short sentences, which is a basic requirement for academic paper writing.

- Where does the number 34.8% in line 75 come from?

- I do not understand the last sentence in line 84-85. "a disconnect between literature and literature?"

- Line 203, human evaluation also @128? Do you mean human evaluate the problem 128 times?

-Line 209, the data contamination is good point, unfortunetaly you did not dive deeper into it.

- Probably you can merge Table 5-7?

- I do not understand the last sentence in line 236-238. What's the relationship between subset and difficulty?

- Related works could be shorter.

**Ethical Concerns:**

["NO or VERY MINOR ethics concerns only"]

**Final Justification:**

Though the authors did not reply my question about the methodology noverty directly, I will still increase my rating to encourage this kind of contribution to the community.

**Limitations:**

yes

**Quality:**

3

**Strengths And Weaknesses:**

## Strengths

1. The authors found systematical errors in a widely-used benchmark in formal reasoning and categorize errors (unprovable problems, incomplete statements, excessive simplifications, etc.).

2. Unlike previous work relying on LLM evaluation, they conduct expert human verification, revealing that reported autoformalization accuracies are inflated (66% vs. 97% claimed by LLMs).

3. miniF2F-v2 provides a higher-quality benchmark with verified formal proofs and corrected statements that better reflect original problem difficulty.

## Weaknesses

1. Apart from the correction of the miniF2F benchmark, i can hardly tell any other contribution of this work. The work is primarily empirical and doesn't provide significant theoretical insights. It is indeed a significant contribution but I think it is more like a major PR to the originial benchmark instead of a well-draft, ready-to-publish academical article.

2. As I know, the miniF2F benchmark has a v2 version (https://github.com/facebookresearch/miniF2F), whereas you did not mention this at all.

3. Human annotation and verification is also a large systemtic project that needs detailed SOP. Pleasy state how do you recruit the annotaters, how to ensure their expertise, and how to make a SOP so that to minimize the inter-person variance during annotation? This is really important as almost all your contributions come from human effort.

4. The work focuses only on Lean language, though miniF2F exists for other formal verification systems (metamath, HOL Light, Isabelle).  Furthermore, the miniF2F-v2 is still limited to the original 488 problems, which is relatively small for comprehensive evaluation of modern AI systems. More problems will make the your work more valuable.

5. Human verification, while more accurate, doesn't scale well compared to automated evaluation methods.  While the paper identifies problems with LLM evaluation, it doesn't propose robust automated alternatives for evaluating autoformalization quality.

6. The paper doesn't fully address potential data contamination issues in some models, though it notes performance differences that suggest this might be occurring.

7. The paper writing needs improvement. It reads not so good. The authors need to spend more time on writing.

---

> ### Author Rebuttal · Authors · 2025-07-31
>
> We thank you for your thoughtful and detailed comments and for your recognition of the strengths in our paper. Below, we address your questions and comments.
>
>
> > Apart from the correction of the miniF2F benchmark, i can hardly tell any other contribution of this work ... It is indeed a significant contribution but I think it is more like a major PR to the originial benchmark ...
> > This is a benchmark study, so why don't you submit to the D&B track?
>
> The NeurIPS CFP for main track lists the topics of interest and the 3rd item on that list is the following:
>
> - **Evaluation (e.g., methodology, meta studies, replicability and validity, human-in-the-loop)**
>
> **We think our paper aligns well with the CFP of the main track.** Although your suggestion of submitting to D&B track is also a sensible possibility, we think the main track is a better fit for our paper given its overall contribution. We have now added experiments on DeepSeek Prover v2 and Goedel Prover v2 extending of our empirical evaluation reporting similar patterns that we believe are insightful for the community.
>
> &nbsp;
>
> ---
>
> > As I know, the miniF2F benchmark has a v2 version (https://github.com/facebookresearch/miniF2F), whereas you did not mention this at all.
>
> Yes, the version that you cited is the first version of miniF2F that included informal statements. The original version released with the miniF2F paper does not include any informal statements. There are several other versions of miniF2F in the literature that have made improvements, each fixing some of the issues in the formal statements. We will discuss this in the paper and cite that specific version as well.
>
> &nbsp;
>
> ---
>
> > Pleasy state how do you recruit the annotaters, how to ensure their expertise, and how to make a SOP so that to minimize the inter-person variance during annotation?
>
> This is a helpful suggestion. We will provide detailed SOP in a new appendix. Our procedures are similar to the one followed by Putnam Bench. We did not recruit any outside annotators. All the work is done by the authors of the manuscript who have been contributing to open source repositories of Lean language.
>
> &nbsp;
>
> ---
>
> > The work focuses only on Lean language, though miniF2F exists for other formal verification systems ... More problems will make the your work more valuable.
>
> We add "Lean" to the title of the paper and the dataset as suggested by reviewer gpAa. Our improvements for the natural language statements and proofs in miniF2F-v2 can be useful for users of Isabelle and Coq and for studies on informal mathematical reasoning. miniF2F-v2 will be a complete benchmark in Lean including human and Lean verified formal/informal statements and formal/informal proofs.
>
> We think extending the miniF2F beyond the 488 problems can be best done in form of a new benchmark with 488 (or even a few thousands) new problems. If we add, let’s say 50 or 200 new problems to miniF2F-v2, that would cause unnecessary confusions comparing it to v1.
>
> &nbsp;
>
> ---
>
> > While the paper identifies problems with LLM evaluation, it doesn't propose robust automated alternatives for evaluating autoformalization quality.
>
> We will discuss this as an important open research question for the entire community. One of the early papers that considers this question for autoformalization is [37]. We think robust automated alternative evaluation methods may be created for autoformalization of specific mathematical topics in the future. However, it may be the case that we need to first develop models that can perform the task of autoformalization, on a given topic, reliably well, and then, use those same models for robust automated evaluation of other models on those specific topics. It is not clear to us whether one can do robust automated evaluation of autoformalization models prior to having models that can reliably autoformalize a given mathematical topic.
>
> Now that we have fixed all the discrepancies between formal and informal statements in miniF2F, one can consider fine-tuning a capable reasoning LLM (LRM), on our miniF2F-v2 and use it as a judge to automatically evaluate new autoformalization models. There are two important caveats here. One is that typical autoformalization models are small (typically 7B) and they are less capable in reasoning compared to LRMs such as o4 and DeepSeek-v3. This might be a limitation for them to judge the correctness of the outputs of other autoformalization models. An extensive and systematic empirical study may shed light on the capabilities of various models for robust automated evaluation in autoformalization. If they demonstrate such capabilities, they will still need to be fine-tuned on high-quality data so that they can achieve high accuracy on a given benchmark. The accuracy of Kimina translator is around 80% on miniF2F based on our human evaluation, still far from being reliably used as an automated judge. The second caveat is that a single natural language statement may have several correct translations in Lean. So, a mere textual comparison of a formal statement produced by a LLM against a ground truth formal statement may not always lead to correct evaluation of accuracy.
>
> This point will be a useful discussion added to our paper which suggests how miniF2F-v2 may be leveraged in future developments.
>
> &nbsp;
>
> ---
>
> > The paper doesn't fully address potential data contamination issues in some models, though it notes performance differences that suggest this might be occurring.
>
> You are right – we will discuss this further in the paper. However, we will refrain from making any claims about data contamination of specific models. It might be plausible that data contamination have adversely affected some of the autoformalization models. However, making such a claim would require an extensive and systematic study of the outputs of these models which is increasingly difficult because the SoTA autoformalization models in the literature have not released their training code and their training data. When model developers are not open about their training sources, studying their models may not lead to adequate insights and clear conclusions. We hope this practice will change in the near future.
>
> There may be other ways to study possible data contamination for these models, but that would require evaluating them not just on miniF2F, but also on other similar statements that are present and absent from the Internet data to measure the sensitivity of these LLMs to modifications in the inputs. Such study on closed-source models would demand a much larger effort by human experts which falls outside the scope of our paper and outside our research interest.
>
> &nbsp;
>
> ---
>
> > The paper writing needs improvement. It reads not so good. The authors need to spend more time on writing.
>
> We will address this feedback and will make sure the final version of the paper reads as smooth as possible.
>
> &nbsp;
>
> --------
>
> > Figure 1 is not necessary as it conveys little information and it occupies a large space in the manuscript.
>
> We appreciate you being open and sharing your view with us. We will consider your suggestion and make a decision after further deliberation.
>
> &nbsp;
>
> ---
>
> > What's the difference between "Translation failure" and "Autoformalization failure"?
>
> They are indeed the same. We will unify these terms and use only one of them in the paper.
>
> &nbsp;
>
> ----
>
> > Can you provide the distribution of systematic errors in the original miniF2F in the main text instead of appendx?
>
> We will address this issue per your suggestion.
>
> &nbsp;
>
> ---
>
> > The sentence spaning line 55-59 is too long, which makes it really difficult to undertand.
>
> Fair point. We will address this and break that sentence.
>
> &nbsp;
>
> ---
>
> > Where does the number 34.8% in line 75 come from?
>
> This number comes from the 4th line in Table 1. Going by the columns from left to right, this number corresponds to the combined accuracy for Herald translator, miniF2F-v2, test, Kimina-ProverDistill-7B.
>
> &nbsp;
>
> ---
>
> > I do not understand the last sentence in line 84-85.
>
> The disconnect that we are referring to in that paragraph is about the mismatches and discrepancies between informal and formal statements in miniF2F-v1. As reported in our paper, a considerable percentage of formal statements are simplified compared to the informal statements. While ATP literature measures the accuracies on the simplified formal statements, the autoformalization literature has built models, such as Kimina Translator, that formalizes the problems without simplification leading to formal statements that are considerably more difficult than the formal statements in miniF2F-v1. As a result, when one evaluates the combines accuracy of a formal reasoning pipeline, the resulting accuracy can be drastically lower than the accuracy of individual modules (ATP and autoformalization). We will clarify this in the final version of the paper.
>
> &nbsp;
>
> ---
>
> > Line 203, human evaluation also @128?
>
> No, our wording in that sentence is too brief which we will revise and clarify. The model is given 1 or 128 attempts to write a proof that passes both the Lean compiler with no error and also the LLM judge – this is the pipeline introduced by Herald at ICLR 2025. The first statement that passes through this system is accepted as the formalization. Once we get the output from Herald's pipeline, we manually evaluate its correctness. In that sentence, we want to compare two cases where the automated pipeline is given 1 vs 128 attempts. Beyond revising the text, we will also replace the term “LLM” with “Herald's automated judge” in Tables 2 and 3.
>
> &nbsp;
>
> ---
>
> Thank you again for your detailed comments and questions. Your feedback improved the clarity of our contribution and how it can be leveraged for advancing the field. In light of these clarification and improvements, we would appreciate it if you consider raising your score. If you have any other questions, please let us know.

---

> > ### Comment · Reviewer_Vohv · 2025-08-05
> > **Reponse to Authors**
> >
> > Thanks for your response. Though the authors did not reply my question about the methodology noverty directly, I will still increase my rating to encourage this kind of contribution to the community.

---

> > > ### Author Response · Authors · 2025-08-06
> > >
> > > We would like to thank the reviewer Vohv for their helpful comments and feedback, and for raising their score.
> > >
> > > To address your last comment about novelty, we will clarify it further with the following statement.
> > >
> > > Our paper, for the first time and through a systematic study, evaluates the performance of a fully automated formal reasoning system consisting of state-of-the-art LLMs for auto-formalization and ATP. Our study indicates that the quality of the miniF2F benchmark can have a drastic positive impact on the accuracy of such systems and that the accuracy of the end-to-end pipeline is significantly lower than that of the individual auto-formalization and ATP modules.
> > >
> > > We also release, for the first time, a fully verified, complete version of miniF2F with formal and informal statements and proofs. While a fully automated system achieves 36.1% accuracy on miniF2F-v1, this increases to 51.2% on miniF2F-v2. Although individual ATP and auto-formalization models report accuracies approaching 100% on miniF2F, our work demonstrates that their combined accuracy remains below 60%, motivating the community to adopt miniF2F-v2 and evaluate their formal reasoning systems in a complete pipeline.
> > >
> > > Another novelty of our work is the extensive human evaluation of auto-formalization outputs, revealing that reported accuracies in the literature are largely inflated (97% reported vs. 66% actual).

---

> > > > ### Comment · Reviewer_Vohv · 2025-08-06
> > > > **Reponse to Authors**
> > > >
> > > > Thanks for your further update. I have no doubt now.

---

### Official Review · Reviewer_mdtW · 2025-06-29

**Clarity:** 4
**Significance:** 4
**Originality:** 2
**Rating:** 4
**Confidence:** 4

**Summary:**

This paper revisits the widely used miniF2F benchmark for evaluating formal reasoning systems. The authors identify critical mismatches and errors between the informal and formal statements in the dataset, which negatively impact evaluations of autoformalization and automated theorem proving (ATP) pipelines. To address this, they propose miniF2F-v2, a revised benchmark with over 300 corrected problems. They also conduct a thorough evaluation of state-of-the-art (SoTA) models in end-to-end theorem proving settings, revealing that the original benchmark significantly overestimated model performance due to flawed ground truth. This paper demonstrates that a cleaner and more faithful benchmark can improve both evaluation fidelity and model development.

**Questions:**

•	What specific types of mathematical statements or constructs most frequently caused failures in autoformalization models?

•	Do you plan to extend miniF2F-v2 corrections to other formal languages beyond Lean to broaden applicability? For example Coq or Isabelle.

•	Can you propose modeling or training strategies to reduce the semantic drift observed in current autoformalizers? The results show significant improvement on v2, but performance is still below 60%. What are the key remaining obstacles?

•	How might data contamination or overfitting to miniF2F v1 affect the validity of prior autoformalization results?

•	Would it be possible to provide automated tools or metrics to detect mismatches in future benchmarks?

•	Would incorporating interactive or verification feedback loops during autoformalization training help close the gap between human and LLM evaluation accuracy?

**Ethical Concerns:**

["NO or VERY MINOR ethics concerns only"]

**Final Justification:**

Thanks to the reviewers for engaging. Some minor concerns are still existing and not resolved so I will keep my marginal accept rating.

**Limitations:**

•	Limited to the Lean formal system, despite miniF2F’s multi-system nature.

•	Does not propose new modeling techniques or training improvements for autoformalization or theorem proving.

•	Benchmark improvements, while critical, are evaluative rather than methodological contributions.

•	Focuses on Olympiad-style problems, so generalization to undergraduate-level or research-level formal mathematics is not explored.

•	Human verification effort is substantial and may not scale for future expansions.

**Paper Formatting Concerns:**

Some figures (especially Figure 1) use small fonts that may be hard to read when printed. Consider enlarging or splitting figures in the final version.

**Quality:**

3

**Strengths And Weaknesses:**

Strengths:

•	Provides a much-needed clean-up of a widely used benchmark.

•	Includes detailed failure analysis and highlights critical flaws in miniF2F-v1.

•	Highlights the limitations of current LLM-based evaluation methods for autoformalization by contrasting them with expert human verification, demonstrating that LLM self-evaluation greatly overestimates accuracy.

•	Provides detailed failure mode analysis, categorizing errors into translation failures, autoformalization failures, semantic drift, and success, guiding future methodological improvements.

•	Releases a valuable and verifiable dataset (miniF2F-v2) with reproducible formal and informal statements.

•	Shows meaningful improvements in end-to-end accuracy using current SoTA models.

Weaknesses:

•	The methodological novelty is limited, focusing primarily on dataset correction and benchmark evaluation rather than proposing new autoformalization or theorem proving algorithms.

•	The study focuses exclusively on the Lean language despite miniF2F’s availability in other formal systems, potentially limiting its broader impact.

•	While the paper presents extensive empirical evaluations, it lacks in-depth analysis of why current autoformalizers fail at specific mathematical constructs or problem categories beyond general semantic drift.

•	No discussion on strategies for improving autoformalization models beyond highlighting benchmark-based discrepancies; readers may expect proposed solutions or training improvements.

•	The human evaluation process, though critical to the claims, lacks detail (e.g., annotator agreement or consistency).

Quality:

The paper is technically sound, with rigorous evaluation methodology, extensive manual verification, and transparent reporting of results and limitations. However, the scope is primarily evaluative rather than methodological.

Clarity:

Overall clear, though the dense presentation of empirical results (e.g. Tables 1-7) could be better summarized in prose to guide the reader to key insights.

Significance:

Highly significant as a benchmark paper, improving data quality for a key dataset in formal reasoning. Its findings on inflated LLM autoformalization evaluations are impactful for the community, though it does not advance modeling techniques itself.

Originality:

The originality lies in the benchmark correction and holistic pipeline evaluation, rather than in algorithmic innovation. However, these contributions are critical for realistic progress in automated formal reasoning.

---

> ### Author Rebuttal · Authors · 2025-07-31
>
> We thank you for your thoughtful and constructive comments and for your recognition of the strengths in our paper. Below, we address your questions and comments.
>
> &nbsp;
>
> --------
>
>
> > The methodological novelty is limited, focusing primarily on dataset correction and benchmark evaluation rather than proposing new autoformalization or theorem proving algorithms.
>
>
> This is a fair point, but we would like to point out the **NeurIPS CFP for the main track** and the third item on its topics of interest:
>
> - **Evaluation (e.g., methodology, meta studies, replicability and validity, human-in-the-loop)**
>
> We think we have made a useful contribution for this item in the CFP.
>
>
> &nbsp;
>
> --------
>
> > The study focuses exclusively on the Lean language despite miniF2F’s availability in other formal systems, potentially limiting its broader impact.
> Do you plan to extend miniF2F-v2 corrections to other formal languages beyond Lean to broaden applicability? For example Coq or Isabelle.
>
> We currently do not have such a plan. Instead we add the word Lean to the title of the paper to make this limitation clear. Our improvements for the natural language statements and proofs in miniF2F-v2-Lean can be useful for users of Isabelle and Coq and also for studies on informal mathematical reasoning. miniF2F-v2 will be a complete benchmark in Lean including human and Lean verified formal/informal statements and formal/informal proofs.
>
>
> &nbsp;
>
> --------
>
> > While the paper presents extensive empirical evaluations, it lacks in-depth analysis of why current autoformalizers fail at specific mathematical constructs or problem categories beyond general semantic drift.
> What specific types of mathematical statements or constructs most frequently caused failures in autoformalization models?
>
>
> We will add additional analysis of failure modes of autoformalization models by categorizing the failures and presenting the accuracies/failures based on the categories. Indeed, there exists some patterns in those failures. For example, Herald consistently makes mistakes in formalizing problems that involve a minimum or maximum bound on a set or a function.
>
> &nbsp;
>
> --------
>
> > No discussion on strategies for improving autoformalization models beyond highlighting benchmark-based discrepancies; readers may expect proposed solutions or training improvements.
>
> This is a great point. We will add this discussion. We think the first step for improving autoformalization models is to adopt better evaluation practices using high quality benchmarks. This is the step we take in this paper. As the next step, we propose creating high quality training data paired in formal and informal language, and removing any discrepancies from the existing training sets which are often closed source. As the validation set of miniF2F is often used for training, the new benchmark may have a direct positive effect for model development, too. Any training sample that has a discrepancy between its formal and informal statements may have an adverse effect on the accuracy of autoformalization models. We will cite references from the literature on LLMs and general ML. The distribution (math topics, number of samples, etc.) of training set may also have a significant effect on the final accuracy of a model trained on the data. The model architecture, its reasoning modules, context length, automated feedback, etc. have also proved to be influential in the final accuracy of LLMs and other ML models. We will mention this in the paper. Extensive ablation studies are, perhaps, currently lacking in the autoformalization literature. We hope such studies will be added to the literature in the near future. Again, **having correct and complete benchmarks will make such ablation studies more informative because if the benchmarks have discrepancies, interpretation of ablation studies will be noisy and foggy.** We hope our work facilitates future developments in this field.
>
> &nbsp;
>
> --------
>
> > The human evaluation process, though critical to the claims, lacks detail (e.g., annotator agreement or consistency).
>
> We will provide an appendix detailing the procedures followed by the human annotator.
>
> &nbsp;
>
> --------
>
> > Can you propose modeling or training strategies to reduce the semantic drift observed in current autoformalizers? The results show significant improvement on v2, but performance is still below 60%. What are the key remaining obstacles?
>
> The remaining obstacles are two fold in our view:
>
> First, we need autoformalizers that can translate with higher accuracy. For the start, such autoformalizers need to be trained on higher quality data. Any training sample which has a discrepancy between the informal and formal statements may have a negative effect on the accuracy of an autoformalizer.
>
> The second fold is about theorem provers. **Our new results which will be added to the paper indicate that, merely on the ATP task, the accuracy of SoTA model, Goedel Prover v2, drops by about 10% when faced with the faithful formal statements that we provide in miniF2F-v2-Lean. This accuracy drops further when Goedel Prover v2 is paired with Kimina translator in the complete formal reasoning pipeline that starts from natural language statements.**
>
> In this work, we remove the barrier of discrepancies and errors in miniF2F and show that the higher quality benchmark has a positive impact on the accuracy of the complete formal reasoning pipeline that proves theorems starting from natural language statements.
>
> &nbsp;
>
> --------
>
> > How might data contamination or overfitting to miniF2F v1 affect the validity of prior autoformalization results?
>
> This is an important point. It might be plausible that data contamination and/or overfitting have adversely affected some of the autoformalization and ATP models. However, making such a claim would require an extensive and systematic study of the outputs of these models which is increasingly difficult because the SoTA autoformalization models in the literature, unfortunately, have not released their code and their training data.  When model developers are not open about their training sources – do not release training data and training code – studying their models may not lead to adequate insights and clear conclusions. We hope this practice will change in the near future.
>
> There may be other ways to study these models, but that would require evaluating them not just on miniF2F, but also on other similar statements that are present and absent from the Internet data, measuring sensitivity of these LLMs to modifications in the problems. Such study would demand a much larger effort by human experts which falls outside the scope of our paper and outside our research interest.
>
> &nbsp;
>
> --------
>
> > Would it be possible to provide automated tools or metrics to detect mismatches in future benchmarks?
>
> We think this is possible using high quality data and also larger models with advanced reasoning capabilities. For example, one can consider training an advanced LLM with reasoning capabilities on miniF2F-v2 test and validation sets, and then, use that LLM merely as a judge to evaluate the accuracy of autoformalization models that are released in the future. This can also be done for other benchmarks such as Putnam Bench. Having high quality data paired in formal and informal language would be a prerequisite for developing such automated system.
>
> &nbsp;
>
> --------
>
> > Would incorporating interactive or verification feedback loops during autoformalization training help close the gap between human and LLM evaluation accuracy?
>
> Yes, this is a great idea for future research, and we believe miniF2F-v2 will be useful for correct evaluation of such study. o4-mini is evidently a general purpose LLM and has not been fine tuned for the task of autoformalization and ATP. Yet, in a previous published paper, using in context learning, we have reported that o4-mini has an impressive capability in incorporating feedback from the Lean compiler. It is likely that using feedback from Lean compiler will also be helpful during fine-tuning with or without RL.
>
> &nbsp;
>
> --------
>
> Thank you again for your detailed comments and questions. Addressing the points that you raised improves the clarity of our paper, and the revised paper will provide better insights about possible ways to leverage the miniF2F-v2-Lean for future work. In light of these clarifications and improvements in the paper, we would appreciate it if you consider raising your score. If you have any additional questions, please let us know.

---

> > ### Comment · Reviewer_mdtW · 2025-08-05
> >
> > Most of my major concerns were not addressed.
> > (1) The authors are not going beyond Lean.
> > (2) "We will add additional analysis of failure modes of autoformalization models by categorizing the failures and presenting the accuracies/failures based on the categories. Indeed, there exists some patterns in those failures. For example, Herald consistently makes mistakes in formalizing problems that involve a minimum or maximum bound on a set or a function." I was hoping this detail would have been provided in the rebuttal.
> > (3) "We will provide an appendix detailing the procedures followed by the human annotator.". I was hoping to get some details in the rebuttal.
> > (4) You could use open models to address the overfitting question.
> >
> > Given this, I am more inclined to reduce my rating.

---

> > > ### Author Response · Authors · 2025-08-06
> > >
> > > We would like to thank reviewer mdtW for their additional feedback. We understand this feedback aims to further enhance our presentation and contribution.
> > >
> > > > The authors are not going beyond Lean.
> > >
> > > The main contribution of this work lies in the complete verification of a widely used benchmark and demonstrating the effect of the quality of this benchmark in a fully automated formal reasoning system.  Formalizing the problem statements in another language, matching translations, and writing complete proofs in formal and informal languages are **non-trivial tasks that takes months to perform, refine, and present to the community**. Constructing and correcting miniF2F in a similar fashion for other languages, such as Isabelle, leads to substantial overhead that goes beyond a simple expansion of results.
> > >
> > > Moreover, evaluating full pipelines in different languages requires extensive GPU hours and significant implementation effort. We have done extensive human and LLM verification and experiments on both versions of the dataset in Lean. We pair SoTA autoformalizer and ATP models to build the complete formal reasoning pipeline. These are the models we have paired in our experiments:
> > >
> > > For autoformalization: HERALD, Kimina Autoformalizer, and o4-mini
> > >
> > > For ATP: DeepSeek v1.5, Goedel Prover v1, Kimina Prover 7B
> > >
> > > Extending beyond Lean4 and verifying state-of-the-art pipelines from the ground up for other languages, such as Isabelle and Coq, would itself be a novel contribution. Since none of the provers support multiple languages, presenting all results in one work would be very challenging, let alone testing every possible combination of translation models and theorem provers.
> > >
> > > Additionally, miniF2F is one of the few cross-language benchmarks. Other major benchmarks in the literature, such as ProofNet, PutnamBench, and LeanDojo, focused exclusively on Lean4. Putnam Bench was later translated to Isabelle several months after its original publication and without presenting any empirical evaluations. Therefore, as other reviewers have noted, we believe it is fair to include “Lean” in the benchmark’s name to better manage readers’ expectations.
> > >
> > >
> > > > "We will add additional analysis of failure modes of autoformalization models by categorizing the failures…" I was hoping this detail would have been provided in the rebuttal.
> > >
> > > We thank the reviewer for their increased interest in translation failure modes. To provide more detail per your suggestion, we conducted a **human review of auto formalizations** of HERALD on the miniF2F-v2-test. To save space in this response, we grouped rare topic mistakes under the “Others” category.
> > >
> > > | Failure cases| Percentage (%) |
> > > |---|---|
> > > | Wrong/missing/incomplete translations| 17.50|
> > > | Type wrong/mismatch| 16.25 |
> > > | Max/Min of sets/functions wrongly formalized | 16.25|
> > > | Incomplete set of assumptions| 12.50|
> > > | Rejected by Lean compiler | 7.50|
> > > | Digits| 7.50 |
> > > | Others| 22.50 |
> > >
> > > Our results provide insights about weakness of this LLM and topics/areas that can possibly be the focus for future improvements.
> > >
> > > However, we would like to note that our paper reports discrepancies in performance within a full-pipeline setting. We do not advocate for any specific auto formalization model or ATP. We will expand presented results to other autoformalization models (Kimina autoformalizer, o4-mini) in the revised manuscript.
> > >
> > > > "We will provide an appendix detailing the procedures followed by the human annotator.". I was hoping to get some details in the rebuttal.
> > >
> > > The annotation procedures mirror those of PutnamBench. Every problem has undergone double verification. We will include the full annotation and verification procedures in the revised manuscript.
> > >
> > > > You could use open models to address the overfitting question.
> > >
> > > We would like to emphasize that we are not suggesting that overfitting is the sole cause of the inflated accuracy reported for auto-formalization models such as HERALD. While overfitting may be an underlying issue for some models, the main challenge the community faces, in our view, is benchmarking and evaluation practices, beginning with the discrepancies and mismatches in the benchmark itself, i.e. miniF2F.
> > >
> > > We focus on two issues towards improving evaluation practices and building complete formal reasoning systems:
> > >
> > > 1. Providing a complete benchmark with 4 components that are fully verified: formal/informal statements and formal/informal proofs.
> > >
> > > 2. A fully automated formal reasoning system that performs both auto formalization and theorem proving in tandem and demonstrating the effect of dataset quality on the accuracy of such a pipeline.
> > >
> > > Our conclusion is not that overfitting is the main issue here. Studying overfitting of some other open source auto formalization models does not relate to the topic and findings of our paper.
> > >
> > > Future studies on overfitting of formal reasoning models will benefit from access to a complete, error-free benchmark.

---

> > > > ### Comment · Reviewer_mdtW · 2025-08-08
> > > >
> > > > Thank you very much to the authors for providing lots of details to rebut.
> > > >
> > > > (1) The argument that going beyond Lean itself is a major contribution is not something I agree with. However, adding Lean in the title is good to be clear of your contribution.
> > > >
> > > > (2) Thank you for providing some more details on this. It would be good to address how these failures impact the final performance and is there any insight you can draw.
> > > >
> > > > (3) You have really not addressed the question of using open models.
> > > >
> > > > Overall, looking at other comments, I think the changes suggested for the paper will improve the paper. I already have a borderline accept and will stay with that rating. Again, thank you to the authors for engaging with me and other reviewers.--

---

### Official Review · Reviewer_ZGEa · 2025-07-01

**Clarity:** 3
**Significance:** 3
**Originality:** 3
**Rating:** 5
**Confidence:** 4

**Summary:**

This paper revises the miniF2F benchmark for evaluating end-to-end mathematical reasoning from natural language to formal proofs. The authors show that prior claims of high component-level accuracy (97% for autoformalization, 69% for theorem proving) do not hold in a full pipeline setting, which achieves only 36% accuracy on the original dataset. They trace this to errors and mismatches in problem formalizations.

The corrected benchmark, miniF2F-v2, improves pipeline accuracy to 57%, though significant challenges remain. The paper also finds that previous studies tend to overestimate formalization correctness, with true accuracy at 66%, not 97%. This work provides a stronger benchmark that helps faithfully evaluate models formal reasoning capabilities.

**Questions:**

On line 156: when you check whether “ the first formal output … remains semantically faithful to the source.”, are these verified by human experts or language models? In other places of the paper (like section 4), it is made clear that human experts evaluation is involved, but I can’t seem to find similar statement in this section.

**Ethical Concerns:**

["NO or VERY MINOR ethics concerns only"]

**Final Justification:**

I am happy with the answers to my questions. I’m keeping my scores.

**Limitations:**

Yes, the authors did mention that the improved benchmark is only for Lean. This is understandable because “patches and fixes” for formal statements are always specific to the target proof engine with their libraries.

That said, the errors in informal statements identified by the authors can possibly be made generic.

**Paper Formatting Concerns:**

The formatting seems proper.

**Quality:**

3

**Strengths And Weaknesses:**

Strength

- The authors inspected in detail the current issues with the MiniF2F benchmark. Errors are well classified into two primary categories and 8 sub-categories. The authors clearly put a significant amount of effort into identifying the errors for over 300 problems by hand. I’m mostly impressed by the corrections in formal statements as some of them would require intensive knowledge about the prover (e.g,, knowing the “right” expressions in Lean to use).
- The demonstration of flawed existing studies is solid. The authors defined a fair workflow and verified suspiciously high numbers against that. The manual evaluation of the quality by Lean experts made the results more compelling.
- The improved accuracy when re-evaluating all the models on the improved benchmark (with the fair pipeline) justify the miniF2F-v2 can serve as a clearer and more demanding benchmark.

Weakness
- My only “complaint” is perhaps on what qualifies a “simplification” of the original problem, as discussed in section 2.2. Not all simplifications are bad — sometimes simplification is even necessary because a problem may not be formalizable at all otherwise due to lack of proper definition/constructions in the existing library.

---

> ### Author Rebuttal · Authors · 2025-07-30
>
> We thank you for your thoughtful and constructive comments and for your recognition of our contribution.
>
> &nbsp;
>
> --------
>
> > My only “complaint” is perhaps on what qualifies a “simplification” of the original problem, as discussed in section 2.2. Not all simplifications are bad — sometimes simplification is even necessary because a problem may not be formalizable at all otherwise due to lack of proper definition/constructions in the existing library.
>
> This is a great point and hard to define in general. But within the context of miniF2F, simplification has a specific meaning in our paper. The simplifications that we are concerned about are the ones that lead to a discrepancy between the informal and formal statements, i.e., the kind of simplifications that leads to deduction of points in mathematical Olympiads or exams, and we have a specific criteria which is the reasoning steps needed in the formal proof. A simplified problem will require fewer reasoning steps in its proof. **We will clarify this further in the paper and provide more examples in the appendix.**
>
> One example of simplification is the case where the original IMO1985P6 asks to prove a statement for all real numbers while the miniF2F-v1 version asks to prove that statement only for nonnegative real numbers. Extending the proof of simplified version to negative real numbers is not very difficult but it requires 30 more lines of Lean code.
>
> **We will add a table to report the length of Lean proofs for the simplified problems in miniF2F-v1 vs the the proof lengths for the de-simplified problems in miniF2F-v2. This will demonstrate a significant decrease in the proof lengths for the simplified problems, a proxy for the difficulty of the problems.**
>
> For the problems in miniF2F, we do not see any case where simplification is necessary due to lack of proper definition/constructions in the existing library, i.e., Mathlib. There are cases where we need new definitions, but such definitions are easy and can usually be done in one line of Lean code. For example, the `Real.rpow` function in Mathlib has a specific definition that deviates from the definition of nth root commonly used in high-school mathematics. This mismatch makes one of the problems in miniF2F unprovable. To address this, we define a new function `Real.qpow` which exactly corresponds to definition of nth root used in high school.
>
>
> &nbsp;
>
> --------
>
>
> > On line 156: when you check whether “ the first formal output … remains semantically faithful to the source.”, are these verified by human experts or language models? In other places of the paper (like section 4), it is made clear that human experts evaluation is involved, but I can’t seem to find similar statement in this section.
>
>
> The part “remains semantically faithful to the source” is verified by human. You are correct about other sections. We will clarify it in section 3, too.
>
> &nbsp;
>
> --------
>
>
> Thank you again for your feedback. The discussions and the table that we will add to the paper based on your feedback improve the clarity of our paper and its contribution. If you have any additional questions, please let us know.

---

> > ### Comment · Reviewer_ZGEa · 2025-08-05
> > **Thank you for the response**
> >
> > Thank you for the response. I appreciate the discussion on simplification and my other questions have also been clarified. I would like to keep my positive assessment of the submission.

---

### Official Review · Reviewer_gpAa · 2025-07-08

**Clarity:** 3
**Significance:** 3
**Originality:** 3
**Rating:** 5
**Confidence:** 4

**Summary:**

This paper revisits the miniF2F benchmark for formal theorem proving and autoformalisation in Lean, uncovering widespread issues in miniF2F-v1, including errors and oversimplifications. The authors introduce miniF2F-v2, a corrected version with over 300 human-verified problem pairs. Their evaluation shows that state-of-the-art models perform far worse in end-to-end pipelines than previously reported—dropping from 97% to around 66% accuracy under human judgment. The work highlights the need for stricter evaluation and better alignment between autoformalisation and theorem proving, offering miniF2F-v2 as a more faithful benchmark for future research.

**Questions:**

- If I understand correctly, all 488 problems have been manually proved in Lean. Given that these formal proofs will now be publicly available online, how do the authors suggest handling or reporting potential data leakage when benchmarking future LLM-based provers that may have seen this data during training?

- Regarding line 179 and Table 3: could you clarify why o4-mini uses a different sampling budget than the other models? Also, which LLM was used to evaluate formalization accuracy in Table 3? Was it o4-mini itself? I’m not entirely clear on why the LLM-verified accuracy is omitted for o4-mini—is it due to concerns about self-evaluation or some other reason? A brief explanation would help.

**Ethical Concerns:**

["NO or VERY MINOR ethics concerns only"]

**Limitations:**

yes

**Paper Formatting Concerns:**

n.a.

**Quality:**

4

**Strengths And Weaknesses:**

Strengths
- The original miniF2F benchmark, though impactful, contained numerous errors and ambiguities. In addition, the existence of multiple unofficial versions maintained by different groups has caused confusion within the community. Given the extensive human effort invested in verification, I believe miniF2F-v2 is a significant and much-needed step forward, and I hope it can serve as the definitive version moving forward.
- The task of generating formal proofs from informal statements is compelling. While it skips the answer-guessing aspect found in benchmarks like PutnamBench, it remains faithful to the spirit of miniF2F.

Weaknesses
- A key feature of the original miniF2F was its support for multiple formal languages. While it’s understandable that miniF2F-v2 focuses solely on Lean, this limitation should be made clearer in the title (e.g., miniF2F-v2-Lean) and abstract to better manage reader expectations and avoid burying this information in the limitations section.

---

> ### Author Rebuttal · Authors · 2025-07-30
>
> Thank you for your thoughtful and constructive feedback and for your recognition of our contribution.
>
>
> &nbsp;
>
> Regarding the limitation of not including Isabelle and Coq as covered by the original miniF2F, we will add the word Lean to the title of the paper to make this limitation completely clear. Thank you for this suggestion.
>
> &nbsp;
>
> --------
>
> > If I understand correctly, all 488 problems have been manually proved in Lean. Given that these formal proofs will now be publicly available online, how do the authors suggest handling or reporting potential data leakage when benchmarking future LLM-based provers that may have seen this data during training?
>
> Yes, your understanding is correct. About data leakage, we can think of two different settings.
>
> Setting 1 is when research papers develop new models or methods (such as training algorithms, etc.) and they want to demonstrate the advantage of their model/method by reporting their accuracy on miniF2F-v2-Lean. In this scenario, we do not have a concrete rule. We think each and every paper can be considered a special case, and the reviewers of each paper will have the best judgement as to what would be scientifically sound. We think after the release of our dataset, if a paper comes forward reporting high accuracy on miniF2F-v2 test without disclosing its training set and releasing a reproducible code, the reviewers of such paper may not be convinced about the technical contribution of the paper. That said, we can imagine a sound paper to openly fine-tune a model directly on the test set of miniF2F in various settings and report the accuracy of resulting models in each setting. We have such studies in the computer vision field that have provided useful insights for the community. Computer vision community has had access to all images of ImageNet and their labels, and through community enforced standards, they have not faced major difficulties in making progress towards better models with better generalization capabilities.
>
> We also envision a line of research to build upon the work of papers such as LeanDojo, train models from scratch on open source data, and demonstrate better and better ATP accuracies starting from natural language statements of miniF2F-v2 and other benchmarks. We think, gradually, the focus of our community will also expand towards other aspects such as the number of tokens used to achieve a high formal reasoning accuracy, especially in the complete formal reasoning pipeline that we study.
>
> Setting 2 is when models are facing new problems such as proving the problems in IMO 2025 during the Olympiad's timeframe. In this case, we think models can be trained on any data with no need to disclose their sources. This is similar to students being allowed to study any material that they desire during their preparation and with no need to report what they have studied. Each Olympiad has its own rules such as time, no access to internet, etc. An AI model should also follow those rules.
>
> &nbsp;
>
> --------
>
> > Regarding line 179 and Table 3: could you clarify why o4-mini uses a different sampling budget than the other models? Also, which LLM was used to evaluate formalization accuracy in Table 3? Was it o4-mini itself? I’m not entirely clear on why the LLM-verified accuracy is omitted for o4-mini—is it due to concerns about self-evaluation or some other reason? A brief explanation would help.
>
> This is a great clarification point. In Table 3, DeepSeek-V3 was used for reproducing the pipeline used by Herald Translator, the autoformalization model with highest accuracy published in ICLR 2025. This is the exact pipeline they used and we deployed their own code released with their paper. They had used DeepSeek-V2.5 which is no longer available via API. So, we used Deepseek-V3 and achieved the same result as reported in Herald’s paper. We will clarify this in Table 3 and the text. Specifically, in Tables 2 and 3, we will replace LLM with “Herlad’s automated judge”.
>
> In our paper, the outputs of Herald and Kimia translator are first evaluated using Herald’s pipeline @1 and @128. The final outputs that come through this pipeline, deemed correct by DeepSeek-V3, are evaluated by human expert. For o4-mini, however, we devised a completely different pipeline which we will make more clear in the paper and the tables.
>
> o4-mini is a general purpose model that is not exactly intended to be used for autoformalization in Lean4. Notably, it often writes code in Lean 3, and as far as we know, its autoformalization accuracy on miniF2F is not reported in the literature. However, o4-mini is a strong reasoning model and it can effectively make use of feedback, e.g., feedback from the Lean compiler.
>
> After some initial experiments, we concluded that it would not make sense to experiment with o4-mini in the same setting as Herald and Kimina translator because its accuracy will be very low while its computational cost will be much higher than the other two models – worst performance at a higher cost would not be interesting to report for a general purpose model that is not intended for autoformalization. Based on this, we created a new setting for o4-mini in which it gets up to 10 rounds of feedback from the Lean compiler to fix the errors in its Lean statements. This new setting improved the accuracy of o4-mini and reduced its cost which was a better use of our computational budget, yet o4-mini’s final accuracy is far below the accuracy of Kimina translator even pass 1 (it comes close to the accuracy of Herald on miniF2F-v1 and exceeds it on miniF2F-v2).
>
> We could have dropped our results on o4-mini to only include the results on the two SoTA autoformalization models. But, we think our results on o4-mini is informative and it may be useful to include in the literature. We will explain this more clearly in the paper which has the potential for future research: reasoning models getting feedback from the Lean compiler to improve their output for autoformalization.
>
>
> &nbsp;
>
> --------
>
>
> Thank you again for your feedback which improves the clarity of our paper. If you have any additional questions, please let us know.

---

### Comment · Area_Chair_49BH · 2025-08-04
**Friendly Reminder to Acknowledge or Update Your Review**

Dear Reviewers,

Thank you for your time and effort in reviewing the submissions and providing valuable feedback to the authors.

If you haven't already done so, we kindly remind you to review the authors' rebuttals and acknowledge them by clicking the "Mandatory Acknowledgement" button at your earliest convenience. This step ensures efficient communication and helps finalize the process smoothly.

We sincerely appreciate your dedication and collaboration.

Best,

Your AC

---

### Decision · Program_Chairs · 2025-09-17

**Decision:**

Accept (poster)

**Comment:**

The paper studies the popular miniF2F benchmark for formal theorem proving. It compares the typical theorem proving task to the full pipeline involved in participating in a math Olympiad, finding that state-of-the-art models perform much worse in an end-to-end pipeline. The paper finds discrepancies between the informal and formal statements in miniF2F, various other errors in miniF2f, and provides a new miniF2F-v2 with the discrepancies corrected.

The paper had several strengths. miniF2F is widely used in AI theorem proving yet has several flaws. The work found interesting errors in informal statements (e.g., unprovable statements) and formal statements (e.g., simplifying the problem). Their resulting improved version should be impactful for the community. In addition, the task of generating formal proofs from informal statements is compelling and interesting to study further, and the paper provides some nice analysis of performance discrepancies between end-to-end proving and proving given a formal statement.

On the other hand, the reviewers highlighted some weaknesses. There is a potential for data leakage, since the proofs are provided with the benchmark. While the authors argued that this can be taken care of based on the experimental methodology used, it remains a fair concern for interpreting the results of some methods tested on the benchmark.

Second, several reviewers noted that the paper only performs analysis on Lean, while the original miniF2F dataset is multi-language. During the discussion the authors acknowledged that they will change the title to make it clear that the paper is about Lean. Given that Lean is one of the main languages considered in AI formal theorem proving, and the paper's contribution was quite substantial in Lean (e.g., providing manually-written proofs), I think this is a sufficient change.

Finally, two reviewers noted that the paper did not provide a methodological or theoretical innovation. The authors clarified that the NeurIPS CFP has "Evaluation (e.g., methodology, meta studies, replicability and validity, human-in-the-loop)". I agree with this and do not view this as a significant concern for this paper.

Overall, the paper presents an interesting analysis and improved version of the widely-used miniF2F benchmark. This represents an interesting contribution that is likely to be impactful. Reviewers' final ratings were all positive, either suggesting accept or borderline accept. Of the borderline accept cases, substantial concerns were addressed in the rebuttal phase. Taking this and the discussion above into account, I recommend acceptance.